**Comparison of Ozone Measurement Methods in Biomass Burning Smoke: An evaluation under**
**field and laboratory conditions**
Russell W. Long[1], Andrew Whitehill[1], Andrew Habel[2], Shawn Urbanski[3], Hannah Halliday[1], Maribel Colón[1], Surender
Kaushik[1], Matthew S. Landis[1]
[1]Center for Environmental Measurement and Modeling, Office of Research and Development, United States Environmental
Protection Agency, Research Triangle Park, North Carolina, United States of America
[2]Jacobs Technology Inc., Research Triangle Park, North Caroline, United States of America
[3]U.S. Forest Service, Rocky Mountain Research Station, Missoula, MT, United States of America
*Correspondence to*: Russell W. Long (long.russell@epa.gov; (919) 541-7744)
**Abstract**
In recent years wildland fires in the United States have had significant impacts on local and regional air
quality and negative human health outcomes. Although the primary health concerns from wildland fires
come from fine particulate matter ($PM_{2.5}$), large increases in ozone ($O_3$) have been observed downwind
of wildland fire plumes (DeBell et al., 2004; Bytnerowicz et  al., 2010; Preisler et al., 2010;Jaffe et al.,
2012; Bytnerowicz et  al., 2013; Jaffe et al., 2013; Lu et al., 2016; Lindaas et al., 2017; McClure and
Jaffe, 2018; Liu et al 2018; Baylon et al., 2018; Buysse et al. 2019). Conditions generated in and around
wildland fire plumes, including the presence of interfering chemical species, can make the accurate
measurement of $O_3$ concentrations using the ultraviolet (UV) photometric method challenging if not
impossible. UV photometric method instruments are prone to interferences by volatile organic
compounds (VOCs) that are present at high concentrations in wildland fire smoke. Four different $O_3$
measurement methodologies were deployed in a mobile sampling platform downwind of active prescribed
grassland fire lines in Kansas and Oregon and during controlled chamber burns at the United States Forest
Service, Rocky Mountain Research Station Fire Sciences Laboratory in Missoula, Montana. We
demonstrate that the Federal Reference Method (FRM) nitric oxide (NO) chemiluminescence monitors
and Federal Equivalent Method (FEM) gas-phase (NO) chemical scrubber UV photometric $O_3$ monitors
are relatively interference-free, even in near-field combustion plumes. In contrast, FEM UV photometric
$O_3$ monitors using solid-phase catalytic scrubbers show positive artifacts that are positively correlated
with carbon monoxide (CO) and total gas phase hydrocarbons (THC), two indicator species of biomass

burning. Of the two catalytic scrubber UV photometric methods evaluated, the instruments that included a Nafion® tube dryer in the sample introduction system had artifacts an order of magnitude smaller than the instrument with no humidity correction. We hypothesize that Nafion®--permeable VOCs (such as aromatic hydrocarbons) could be a significant source of interference for catalytic scrubber UV photometric $O_3$ monitors, and that the inclusion of a Nafion® tube dryer assists with the mitigation of these interferences. The chemiluminescence FRM method is highly recommended for accurate measurements of $O_3$ in wildland fire plume studies and at regulatory ambient monitoring sites frequently impacted by wildland fire smoke.

## 1 Introduction

Ground-level ozone ($O_3$) is a secondary air pollutant generated from the photochemical interactions of nitrogen oxides ($NO_x$) and volatile organic compounds (VOCs). The most robust methods for $O_3$ measurements are based on chemiluminescence reactions with ethylene (ET-CL, for ethylene chemiluminescence) or nitric oxide (NO-CL, for nitric oxide chemiluminescence) (Long et al., 2014). The overall reaction mechanism for ET-CL generally proceeds as detailed in Eqs. (1-2):

$$C_2H_4 + O_3 \rightarrow H_2CO^* + \text{Other products}, \tag{1}$$
$$H_2CO^* \rightarrow H_2CO + h\nu \tag{2}$$

The reaction generates electronically-activated formaldehyde ($H_2CO^*$) which luminesces in the high ultraviolet (UV) to visible portion of the spectrum (380 nm - 550 nm) and vibrationally-activated hydroxide ions which luminesce in the visible light to the low infrared (IR) portion of the spectrum (550 nm - 800 nm). The number of photons emitted during the reaction is directly proportional to the amount of $O_3$ present and are counted by a photomultiplier tube (PMT), with its response centered at 440 nm, then the count is converted to $O_3$ concentration. The ET-CL method requires a constant supply of ethylene for continuous operation. NO-chemiluminescence analyzers measure $O_3$ concentrations using the principle that the dry, gas-phase reaction between NO and $O_3$ generates nitrogen dioxide in an electronically excited state ($NO_2^*$), and oxygen ($O_2$) (Ollison et. al., 2013; Boylan et.al., 2014). As each

unstable, $NO_2^*$ molecule returns to a lower energy state ($NO_2$), it emits a photon (hv). The reaction causes
luminescence in a broadband spectrum ranging from visible light to infrared light (approximately 590 nm
– 2800 nm). The two-step gas-phase reaction proceeds as detailed in Eqs. (3-4):

$$NO + O_3 \rightarrow NO_2^* + O_2, \qquad (3)$$

$$NO_2^* \rightarrow NO_2 + hv. \qquad (4)$$


The ET-CL method is no longer used nor produced commercially and has been replaced by the NO-CL
method. Similar to the ET-CL method, the NO-CL method requires a constant supply of gas, in this case
NO, for continuous operation. Both the ET-CL and NO-CL methods are subjet to slight interfernces by
water vapor. Howver, these potential interfenrces can be elimitated throught the use of Nafion based drier
or equivalent sample water vapor treatment system. The ET-CL method was promulgated as the Federal
Reference Method (FRM) for measuring $O_3$ in the atmosphere in 1971 and the NO-CL method
promulgated as the FRM in 2015 (U.S. EPA, 2015).

While the chemiluminescence method for measuring $O_3$ is technically robust and free of analytical
artifacts (Long et al., 2014), it is not widely used in the United States. Instead, Federal Equivalent Methods
(FEM) based upon UV photometry are employed at the majority of $O_3$ regulatory monitoring locations.
According to July 2020 data from the United States Environmental Protection Agency (EPA) Air Quality
System (AQS) database, the UV photometric method represents 99% of the roughly 1200 instruments
deployed in network monitoring for $O_3$ National Ambient Air Quality Standard (NAAQS) attainment.
UV photometric methods for $O_3$ are generally considered easier to deploy, operate, and in most cases do
not require external compressed gasses for operation. UV photometric analyzers determine $O_3$
concentrations by quantitatively measuring the attenuation of light due to absorption by $O_3$ present in an
absorption cell at the specific wavelength of 254 nm (Parrish et al., 2000; Williams et al., 2006). The $O_3$
concentration is determined through a two-step process in which the light intensity passing through the
sample air (I) is compared with the light intensity passing through similar sample air from which all $O_3$
is first removed ($I_0$). The ratio of these two light intensity values ($I/I_0$) provides the measure of the light
absorbed at 254 nm, and the $O_3$ concentration in the sample is then determined through the use of the
Beer-Lambert Law as given in Eq. (5):

$$I/I_0 = e^{-KLC} \ (C = 1/KL \ \ln \ [I/I_0]); \tag{5}$$

where L is the length of the absorption cell (cm), C is the $O_3$ concentration (ppm), and K is the absorption
cross section of $O_3$ at 254 nm at standard atmospheric temperature and pressure conditions (308 atm$^{-1}$ cm$^{-1}$
). Photometric monitors generally use mercury vapor lamps as the UV light source, with optical filters
to attenuate lamp output at other than the 254 nm wavelength.

Air for the reference cell measurement ($I_0$) is typically obtained by passing the ambient air sample stream
through a catalytic scrubber containing manganese dioxide ($MnO_2$), hopcalite (a mixture of Cu, Mn, and
Ag oxides), heated silver wool, or another solid state material to 'scrub' only $O_3$ from the sample air while
preserving all other substances in the sample air that potentially absorb at 254 nm (e.g., elemental gaseous
mercury [$Hg^0$], hydrogen, sulfide [$H_2S$], VOCs) so that their effects are cancelled in the differential $I/I_0$
measurement. The integrity of the $O_3$ reference scrubber is critical and may allow measurement
interferences if it does not perform adequately. Similarly, any tendency of the scrubber to fail to
effectively remove all $O_3$ from the reference sample will result in a measurement bias. In addition to $O_3$,
catalytic scrubbers have been shown to remove UV-active VOCs (Kleindienst et al., 1993), creating the
potential for positive artifacts in $O_3$ measurements when the efficiency of this VOC removal is impacted.

Although FEM designated UV photometric instruments are accurate under most ambient conditions,
locations with high VOC concentrations can produce significant analytical artifacts. Smoke plume
impacted locations and measurements downwind from wildland fires are a particular concern; $O_3$
measurements of up to 320 ppb were observed in a smoke plume in western Oregon using a Dasibi
1003AH UV photometric $O_3$ monitor (Huntzicker and Johnson, 1979), which also showed a correlation
between apparent $O_3$ and aerosol concentrations (**$b_{scat}$**, a combustion plume indicator in this case). $O_3$
measurements from UV photometric instruments exceeding 1500 ppb at night (22:00-05:00) were
observed in Fort McMurray, Alberta during smoke impacts from the 2016 Horse River Fire, which were
positively correlated with NO and non-methane hydrocarbons (Landis et al., 2018). Follow-up pyrolysis
experiments demonstrated that ET-CL instruments do not show a similar response to biomass burning
smoke (Huntzicker and Johnson, 1979). Photochemical chamber experiments comparing the $O_3$ response
of UV (Dasibi Model 1003AH, Dasibi Model 1008AH, and Thermo Model 49) and ET-CL (Bendix
Model 8002 and Monitor Labs Model 8410) mixtures show negligible differences for irradiated
paraffin/NOx and olefin/NOx mixtures, but do show a positive UV interference in mixtures with toluene
and other aromatics present (Kleindienst et al., 1993). Laboratory studies comparing the response of UV
(Thermo Model 49, Horiba APOA-370, and 2B Tech Model 202) and ET-CL (Bendix) instruments
showed a positive interference for o-nitrophenol, naphthalene, and p-tolualdehyde for the UV instruments
but not the ET-CL instruments (Grosjean and Harrison, 1985; Spicer et al., 2010). Additionally, during
the Mexico City Metropolitan Area field campaign (MCMA-2003) a mobile laboratory using an FEM
designated UV photometric $O_3$ monitor (unheated $MnO_2$ scrubber, Thermo 49 series) showed a large
positive $O_3$ interference (~400 ppb) associated with $PM_{2.5}$ and polyaromatic hydrocarbons (PAHs) when
following some diesel vehicles (Dunlea et al., 2006). Although not compared to a chemiluminescence
instrument, those high $O_3$ values are unlikely real considering the high concurrent NO concentrations (in
some cases, >1000 ppb). The authors of this study attributed the interference to fine particles, based on
the correlation with $PM_{2.5}$ and the lack of a correlation with gas-phase organic species measured by the
proton transfer reaction-mass spectrometer (PTR-MS, Dunlea et al., 2006).

In addition to interferences from the presence of aromatic VOCs and semi-volatile PAHs, water vapor
(relative humidity) issues have also been observed with older generation FRM and FEM designated
chemiluminescence and UV photometric $O_3$ instruments, respectively (Kleindienst et al., 1993; Leston et
al., 2005;Wilson and Birks, 2006). As such, Nafion® tube dryers are regularly incorporated into some
newer generation chemiluminescence and UV photometric $O_3$ monitors in an attempt to mitigate the
humidity related measurement artifacts.

A recently introduced variation of the UV photometric method, known as the "scrubberless" UV
photometric (SL-UV) method (Ollison et al., 2013), specifies removal of $O_3$ from the sample air for the
reference by a gas-phase reaction with NO rather than using a conventional solid state catalytic scrubber.
The NO gas phase chemical scrubber reacts with $O_3$ much faster and more selectively than with other
potential interfering compounds and is very effective at removing the $O_3$ without affecting other
interfering compounds that may be present in ambient air. The differential UV measurement can then
effectively reduce interferences to an insignificant level. Similar to NO-CL, the SL-UV method requires
a continuous supply of compressed NO or nitrous oxide ($N_2O$) (which the instrument converts to NO) to
serve as the scrubber gas.

In this study, we investigate UV photometric FEM instrument $O_3$ measurement interferences in fresh
biomass burning smoke plumes from prescribed grassland fires and during controlled burn experiments
in a large scale combustion chamber. We directly compare NO-CL FRM $O_3$ measurements to several
FEM designated UV photometric technologies, including a gas-phase scrubber and catalytic scrubbers
with and without Nafion® tube dryer systems. Based on the results from the measurements, we assess the
magnitude of the observed artifacts for different technologies and under various smoke conditions and
provide suggestions for potential mitigation of the interferences.

## 2. Methods

### 2.1 Overview of Methods Evaluated

In this study we compared the measurement results from six different commercially available FRM/FEM
designated $O_3$ instruments operated in ambient or chamber generated biomass burning smoke. All
instruments were operated according to their FRM or FEM designation. The six instruments differed by
measurement principle (chemiluminescence *versus* UV photometric), and by sample treatment
configuration (scrubber material, presence of dryer, etc.). For interference free $O_3$ measurements we
utilized the newly designated FRM NO-CL method (U.S. EPA, 2015). For the UV photometric methods,

we compared both catalytic scrubber and "scrubberless" (gas phase chemical scrubber) technologies, with the "scrubberless" monitor using a NO chemical scrubber. Finally, within the catalytic scrubber UV photometric category, we compared instruments with and without Nafion tube dryer systems. The operation principle and designations (FRM vs FEM) for the analyzers under investigation are summarized in Table 1 and described in Sections 2.1.1-2.1.4. These analyzers were operated immediately downwind of fresh biomass burning plumes during eight days of prescribed fires in grassland ecosystems in Oregon and Kansas and during laboratory-based studies at the U.S. Forest Service's (USFS) combustion facility at the Fire Sciences Laboratory (FSL) in Missoula, Montana. The grassland fire fuels consisted primarily of mixed native prairie tall grass of varying moisture content. Seven of the eight days of prescribed burning were conducted in the Tallgrass Prairie ecosystem of central Kansas (four days in March of 2017 and three days in November of 2017). The additional burn day was conducted at the Sycan Marsh in central Oregon (October of 2017). Laboratory based chamber burns at the FSL were conducted during April 2018 and again during April 2019. Fuels for the laboratory based chamber burns consisted of ponderosa pine needles and fine woody debris. Details of the individual studies are provided in Sections 2.2-2.6.

**Table 1: Ozone measurement methods investigated.**

| Name | Manufacturer | Model | Method | Scrubber | Cells | Humidity Correction | Deployment[a] |
|------|--------------|-------|--------|----------|-------|---------------------|---------------|
| **U.S. EPA Federal Reference Methods (FRM)** | | | | | | | |
| NO-CL | Teledyne API | T-265 | CL (NO) | N/A | 1 | Nafion®-based (dryer) | K1, S, K2, T, M1, M2 |
| **U.S. EPA Federal equivalent methods (FEM)** | | | | | | | |
| UV-C | Thermo Scientific | 49i | UV (254 nm) | Catalyst (MnO$_2$) | 2 | None | K1, S, K2, T, M1, M2 |
| UV-C-H | 2B Technologies | 205 | UV (254 nm) | Catalyst (Hopcalite) | 2 | Nafion®-based (equilibration) | K1, S, K2, T, M1 |
| SL-UV | 2B Technologies | 211 | UV (254 nm) | Gas chemical (NO) | 2 | Nafion®-based (equilibration) | K1, M1, M2 |
| UV-G | 2B Technologies | 211-G | UV (254 nm) | Heated graphite | 2 | Nafion®-based (equilibration) | M2 |

[a]K1-Konza Prairie March 2017; S-Sycan Marsh, October 2017; K2-Konza Prairie November 2017; T-Tallgrass Prairie November 2017; M1-Missoula chamber April 2018; M2-Missoula chamber April 2019.

### 2.1.1 NO Chemiluminescence

The FRM $O_3$ measurement method was the Teledyne API (San Diego, CA, USA) Model T265 Chemiluminescence Monitor (TAPI T265), which utilizes a NO-CL measurement principle. These NO-CL $O_3$ analyzers have been shown to be free of interferences (Long et al. 2014) , and have been used as a reference method in other $O_3$ comparison studies (Williams et al., 2006; Landis et al., 2020). Although there is a known water vapor interference with chemiluminescence technology (Kleindienst et al., 1993), the TAPI T265 uses a Nafion® tube dryer system to remove water vapor from the air prior to making the measurement, thus eliminating any humidity-related effects. Like the ET-CL technologies (Kleindienst et al., 1993), the NO-CL analyzers have no documented VOC interferences. Manufatcurer provided performance specifications for the NO-CL based TAPI T265 are given in Table S1.

### 2.1.2  Catalytic Scrubber UV Photometric

For this study the UV photometric method with no humidity correction was represented by the Thermo Scientific (Franklin, MA, USA) Model 49i (Thermo 49i), which is a dual cell instrument with a manganese oxide ($MnO_2$) catalytic scrubber, referred to as UV-C. Nafion®-based humidity systems or dryers have been employed within photometric $O_3$ monitors with catalytic scrubbers before the measurement cell, offering a reduction in relative humidity interferences and artifacts (Wilson and Birks, 2006). Manufacturer provided performance specifications for the UV-C based Thermo 49i are given in Table S1.

The UV photometric with a Nafion® humidity conditioning system was represented in this study by a 2B Technologies (Boulder, CO, USA) Model 205 (2B 205) $O_3$ monitor. The 2B 205 utilized a dual-cell design where sample air and scrubbed air are measured simultaneously. The 2B 205 uses a Hopcalite ($CuO/MnO_2$) catalytic scrubber to remove $O_3$ from the reference stream. This instrument will be referred to as UV-C-H. Manufacturer provided performance specifications for the UV-C-H based 2B 205 are given in Table S1.

### 2.1.3 Scrubberless UV Photometric

For comparison with the NO-CL, UV-C and UV-C-H methodologies, a "scrubberless" UV (SL-UV) photometric analyzer with a gas-phase (NO) chemical scrubber was employed (Ollison et al., 2013; Johnson et al., 2014). The addition of NO gas to the reference stream selectively scrubs $O_3$ while not significantly affecting interfering VOC species, resulting in an interference free $O_3$ determination. Inclusion of this instrument into the study allows evaluation of the impact of the UV method in general (as compared with chemiluminescence) versus the influence of specific scrubber technologies. The SL-UV method is represented by the 2B Technologies Model 211 "Scrubberless" Ozone Monitor (2B 211). The Model 2B 211 requires a continuous supply of compressed NO or nitrous oxide ($N_2O$) (which the instrument converts to NO). The SL-UV method also utilizes a Nafion®-based sample humidity conditioning system to eliminate any humidity effects. The SL-UV instrument was not used in the October or November 2017 burns due to the lack of the necessary reagent gas (nitrous oxide, $N_2O$) needed to run the instrument. Manufacturer provided performance specifications for the Sl-UV based 2B 211 are given in Table S1.

### 2.1.4 Heated Graphite Scrubber UV Photometric

During the final phase of laboratory-based burning a 2B Technologies Model 211-G UV photometric analyzer (2B 211-G) was operated for comparison to the monitors detailed in Sections 2.1.1-2.1.3. The 2B 211-G differs from the 2B 211 in that it employs a heated graphite scrubber to remove $O_3$ from the reference sample stream ($I_0$) (Turnipseed et al., 2017). The 2B 211-G utilizes the same Nafion®-based sample humidity conditioning system as employed in the 2B 211. For the purposes of this manuscript the UV photmetric method employing the heated graphite scrubber be refered to as UV-G. Manufacturer provided performance specifications for the UV-G based 2B 211-G are given in Table S1.

## 2.2 Prescribed Fire Burn Mobile Sampling Platform

During the prescribed fire grass burns, all study instrumentation (analyzers, data acquisition systems, and peripheral systems) were mounted in portable instrument racks and installed inside an enclosed EPA 4x4 vehicle (Whitehill et al., 2019). The instruments were connected via perfluoroalkoxy alkane (PFA) Teflon® tubing (0.64 cm diameter) to PFA Teflon® filter packs loaded with 47 mm, 5 micron pore size pressure drop equivalent Millipore (Burlington, MA, USA) Omnipore® polytetrafluoroethylene (PTFE) filter membranes which were (i) mounted to a rooftop sampling platform during Spring 2017 sampling, or (ii) connected to a cross-linked Teflon®-coated high flow manifold mounted on the inside roof of the truck compartment during Fall 2017 sampling. The truck was positioned downwind of active biomass burning plumes, usually within meters to hundreds of meters of the active fire line, and positioned so that the trailer was downwind of the sample inlets (to avoid interferences from generator exhaust). In addition to the $O_3$ analyzers under investigation, additional monitors were also operated for the determination of carbon monoxide (CO), NO, $NO_2$, total oxides of nitrogen ($NO_x=NO+NO_2$), and total hydrocarbons (THC, to approximate VOC concentrations). The operation principle and designation (FRM vs FEM) information for the additional analyzers deployed in this study are summarized in Table 2. Data from all instruments was recorded on a Envidas Ultimate data acquisition system.

**Table 2: Additional measurement methods operated during the present study.**

| Pollutant | Manufacturer | Model | Method | FRM/FEM | Deployment[f] |
|---|---|---|---|---|---|
| CO | Teledyne API | 48C | NDIR[a] | FRM | K1, S, K2, T, M1, M2 |
| $NO_2$ | Teledyne API | T500U | CAPS[b] | FEM | K1, S, K2, T, M1, M2 |
| NO, $NO_2$, NOx | Thermo Scientific | 42C | CL ($O_3$)[c] | FRM | K1, K2, T, M1 |
| NO, $NO_2$, NOx | Teledyne API | T200/T201[e] | CL ($O_3$) | FRM | M1, M2 |
| THC | Thermo Scientific | 51i | FID[d] | NA | K2, T, M1, M2 |

[a]Non-Dispersive Infrared Absorption
[b]Cavity Attenuated Phase Shift
[c]Ozone Chemiluminescence
[d]Flame Ionization Detector
[e]The Teledyne API Model T201 is not a designated FRM or FEM however it employs the same operating principle as the FRM designated model T200
[f]K1-Konza Prairie March 2017; S-Sycan Marsh October 2017; K2-Konza Prairie November 2017; T-Tallgrass Prairie November 2017; M1-Missoula chamber April 2018; M2-Missoula chamber April 2019.

All instruments were calibrated with multipoint calibrations before and after each sampling day. All pre- and post-calibrations met our quality performance objectives of +/- 10% and linearity of $r^2 \geq 0.99$. For the $O_3$ analyzers under investigation, field and laboratory calibrations were performed using a Teledyne API Model T700U Dynamic Dilution Calibrator with a NIST traceble $O_3$ photometer and $O_3$ generation system. Zero air for the calibrator was supplied by a Teledyne API Model T701H Zero Air Generator. Calibrations for NO, $NO_2$, $NO_x$ and CO were performed using the same calibrator and zero air generator utilizing a certified EPA protocol tri-blend (CO, NO, $SO_2$) gas cylinder (Airgas). Per the manfactuerer provided operators manual, calibrations for THC were performed using the T700U calibrator and a certified EPA methane/propane gas cylinder (Airgas). FID response factors for organic compounds can vary significantly based upon factors such as carbon number and compound class (Tong and Karasek 1984). The carbon numbers for methane and propane vary by a factor of three and the FID response factors for those compounds may also vary by a similar amount. In addition, the complex mixture of hydrocarbons found in smoke will have large variations in carbon number and FID response factors. As such, the results obtained with the THC analyzer are an approximation of THC (and VOC) concentrations in smoke. In addition, for THC calibrations, the T701H zero air generator was replaced with scientific grade zero air compressed gas cylinders (Airgas).

## 2.3 Kansas Prescribed Burns, March 2017

Biomass burning plumes were sampled during four days of prescribed burns (March 15-17, 2017 and March 20, 2017) on the Konza Prairie Long Term Ecological Research (LTER) site outside of Manhattan, Kansas. The fuels for this series of burns consisted of mixed native prairie tall grass of varying moisture content. Over the four-day period, a total of 13 burns were conducted and sampled.

## 2.4 Oregon Prescribed Burns, October 2017

A single 10-hour day of prescribed grassland burning was measured at the Sycan Marsh Preserve in central Oregon on October 11, 2017. Fuels for the Sycan Marsh burn consisted of mixed native prairie tall grass of varying moisture content.

## 2.5 Kansas Prescribed Burns, November 2017

Biomass burning plumes were sampled during a single day of prescribed burning (November 10, 2017) on the Konza Prairie LTER site outside of Manhattan, Kansas and on two additional days (November 13, 2017 and November 15, 2017) at the Tall Grass Prairie National Preserve outside Strong City, Kansas. Fuels for the November 2017 burns consisted of mixed native prairie tall grass of varying moisture content. During the November 10 sampling, two separate burns were conducted. Three burns were conducted over the two days at Tallgrass Prairie.

## 2.6 USFS Missoula Burn Chamber Burns 2018, 2019

Laboratory-based studies were performed at the U.S. Forest Service's combustion testing facility at the FSL in Missoula, Montana by EPA and USFS personnel. These static chamber burns occurred in the spring of 2018 (April 16-24, 2018; 33 burns; Landis et al., 2020) and again in the spring of 2019 (April 15-26, 2019; 31 burns). The main combustion chamber is a square room with internal dimensions 12.4 x 12.4 x 19.6 m high and a total volume of 3000 $m^3$ and has been described previously (Bertschi et al., 2003; Christian et al., 2004; Yokelson et al., 1996; Landis et al., 2020). During the combustion chamber studies, the facility was fitted with identical instrumentation racks, calibration systems, systems for sampling of test atmosphere, and data acquisition systems, as those described in Section 2.2. All instrumentation were housed in an observation room immediately adjacent to the combustion chamber with PFA inlet lines extending through the wall into the chamber. All inlet lines contained an identical filter pack/filter assembly described in Section 2.2 to protect inlet lines and the analyzers from particulate contamination. Fuel beds consisting of ponderosa pine needles and mixed woody debris were prepared

and placed in the middle of chamber. The amount and moisture content of the fuels were varied to generate
different flaming/smoldering conditions during the burns. During the chamber burns the combustion room
was sealed and the fuel bed was ignited. Two large circulations fans on the chamber walls and one on the
ceiling facilitated mixing and assured homogeneous conditions during the burn periods (Landis et al.,
2020). In general, chamber RH values were below 50% facilitating dry burning condition.

## 3 Results and Discussion

### 3.1 Results from Ozone Measurements in Prescribed Grassland Fire Plumes

$O_3$ measurement results from the Oregon and Kansas prescribed grassland fires studies are shown as the
difference between the FEM and FRM in Fig. 1 and 1-minute average time series plots for the studies are
presented in Supplementary Figs. S1-S3. There were significant differences in the measurement results
obtained from the different $O_3$ monitors operated during the prescribed fires. The UV-C instrument
(Thermo 49i) consistently showed large increases in $O_3$ concentration readings in fresh biomass burning
plumes, with measurements exceeding the FRM measurement by 2-3 ppm. The $O_3$ exceedances were
generally correlated in time with CO and THC (biomass burning indicators) and $NO_2$. These correlations
will be discussed separately. The UV-C-H instrument (2B 205) also showed increased readings in smoke
plumes (also correlated with CO, THC, and $NO_2$), but with absolute measurements roughly an order of
magnitude smaller than the UV-C instruments. The NO-CL (T265) instrument results showed the
opposite behavior, with reductions in $O_3$ readings inversely correlated with increases in $NO_2$
concentrations, as expected from general $O_3$ titration by NO ($NO + O_3 \rightarrow NO_2 + O_2$). For the March 2017
measurements the SL-UV instrument (2B 211) produced readings roughly comparable with the NO-CL
monitor, but with substantially more noise on a minute-to-minute timescale. The "in plume" average $O_3$
concentrations from the four prescribed grassland burning periods are shown in Fig. 2. For the purposes
of this comparison, CO measurements were used as an indicator of when sampling occurred "in plume."
In addition, ambient RH values were generally belwo 50% indicating that the spring and fall 2017
prescribed burns were cunducted under dry conditions.

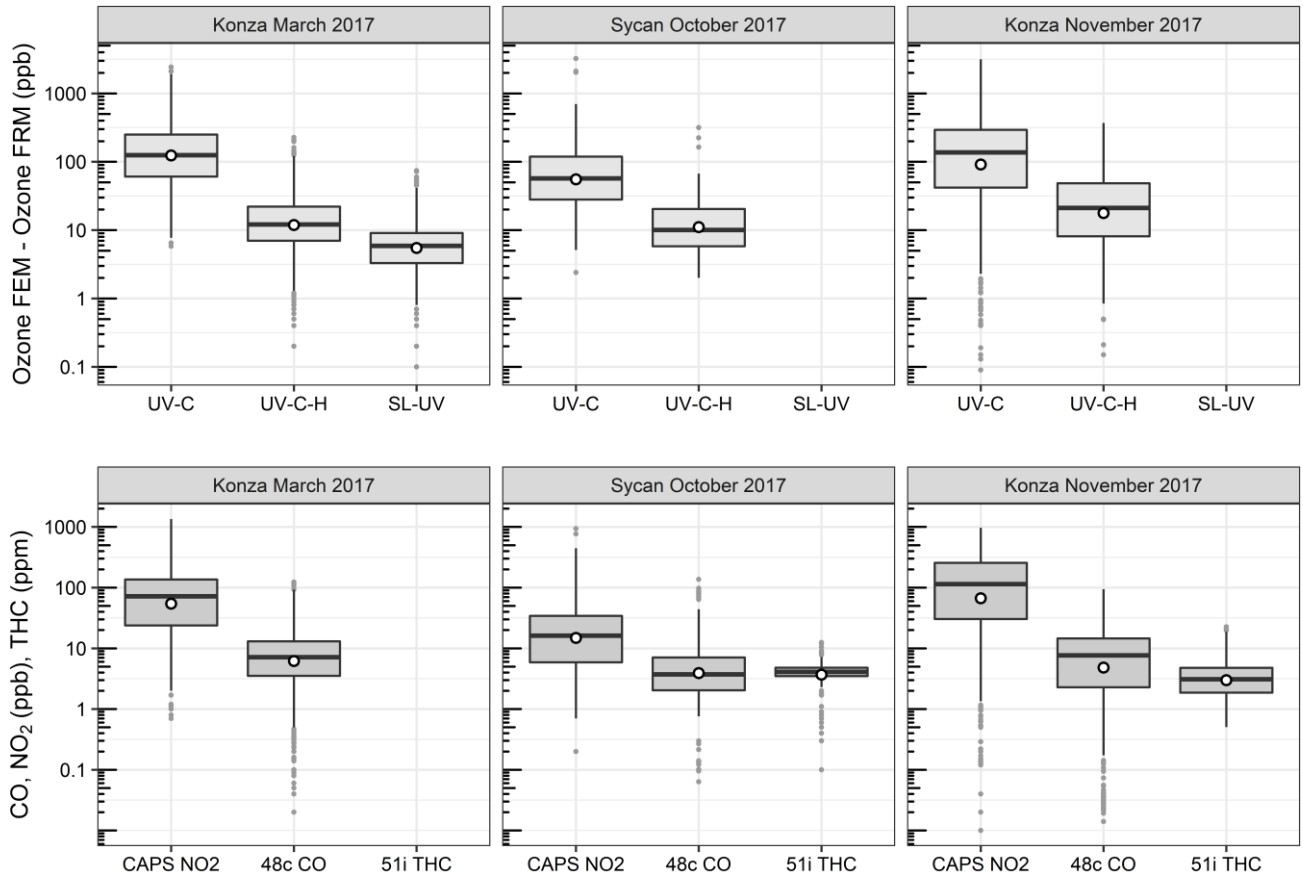


**Figure 1.** Ozone concentration differences between FEM instruments and the FRM instrument (FEM-FRM), and the measured $NO_2$, CO, and total hydrocarbons (THCs) during the three 2017 wildfire deployments. All measurements included are within-smoke only measurements, and show both the elevated smoke tracers ($NO_2$, CO, THC), and the persistent elevation of the FEM $O_3$ measurements. The box and whisker plots indicate the 25th, 50th, and 75th quartiles, with the whiskers extending to 1.5 times the inner quartile range. The open dots indicate the mean values for each instrument within smoke.

346

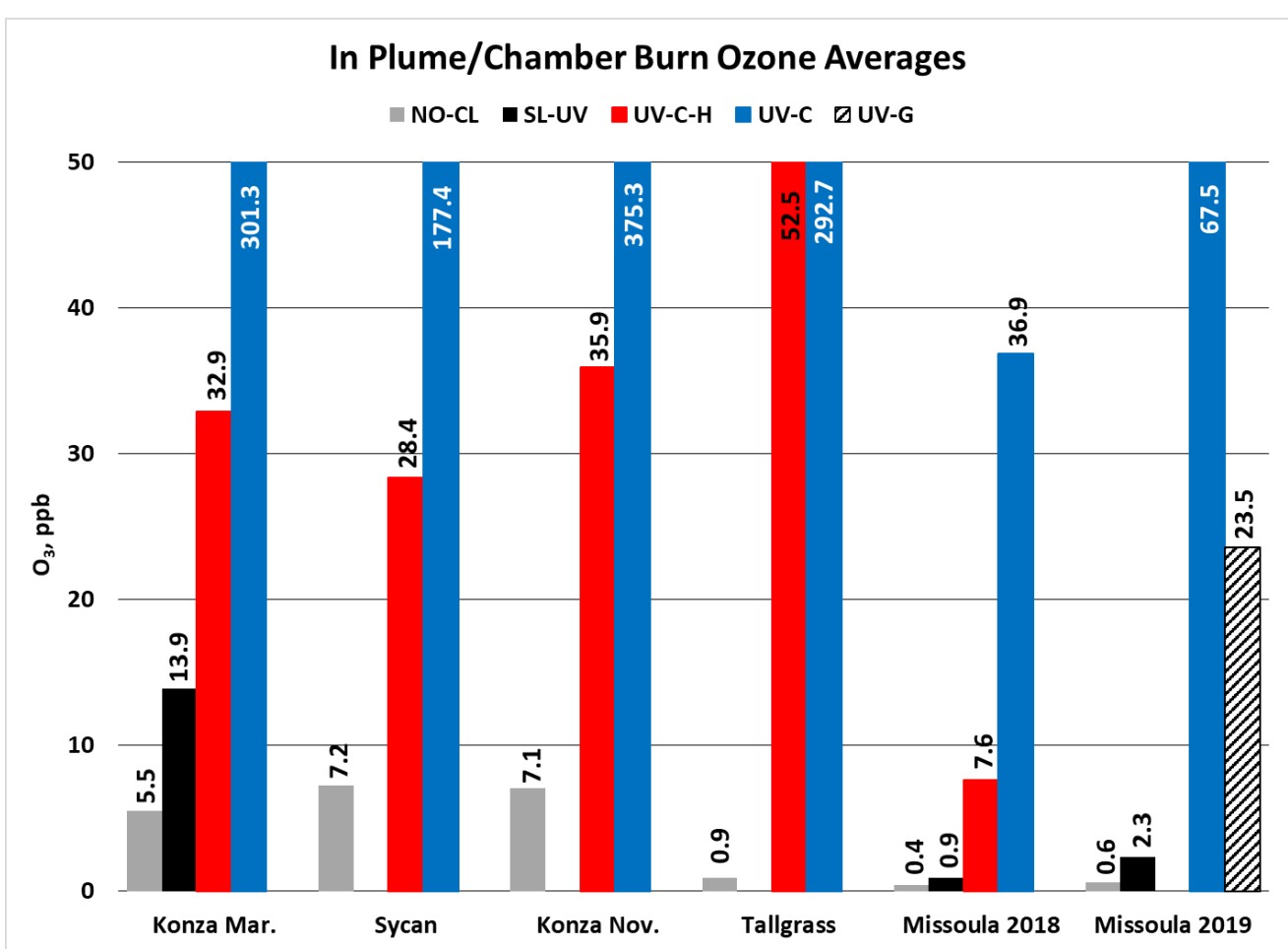

347

**Figure 2.** In plume O$_3$ concentration averages from the 2017 prescribed grassland burns and the 2018 and 2019 Missoula chamber burns. For the 2017 grassland burning periods, CO concentration results ($\geq 1$ ppm) were used as an indicator of when "in-smoke" sampling was occurring.

## 3.2 Results from Ozone Measurements in USFS Chamber Burns

O$_3$ measurement results from the 2018 and 2019 USFS chamber burn studies are shown in Fig. 3. Time series plots of the chamber Study data are included in Supplementary Figs. S4 and S5. Figure 4 gives a more detailed view of UV-C and NO-CL O$_3$ results (two days from 2018 and one day from 2019) during the chamber burns. In contrast to the prescribed grassland burns, the Missoula chamber burns employed differing fuel types (ponderosa pine needles and fine woody debris) that are more typical of fuels

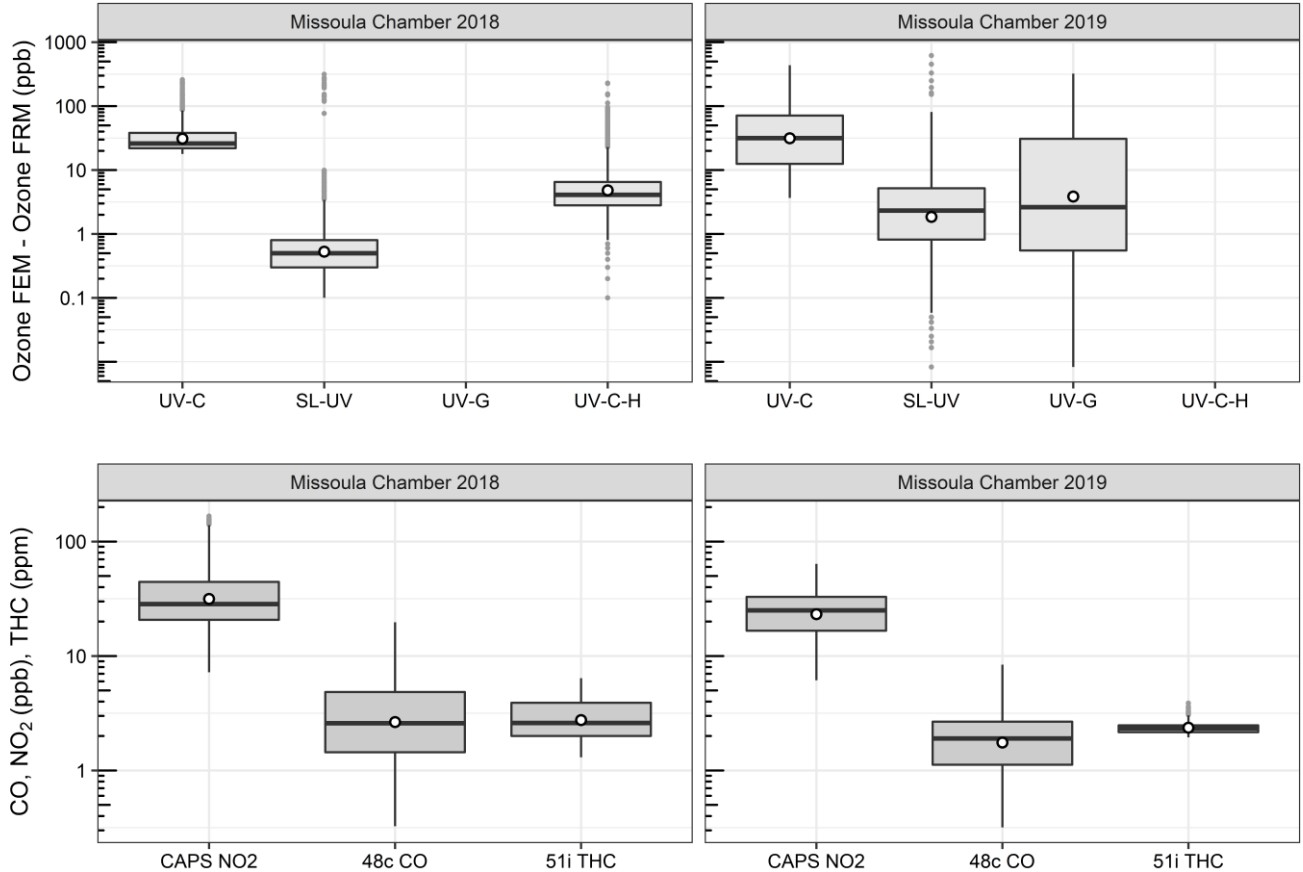

**Figure 3.** Differences between the FEM and FRM instrument concentrations (FEM-FRM), and  $NO_2$, CO, and total hydrocarbons (THCs) concentrations during the 2018 and 2019 Missoula chamber studies. All measurements included are within-smoke only measurements, and show both the elevated smoke tracers ($NO_2$, CO, THC), and the persistent elevation of the FEM $O_3$ measurements compared to the FRM. The box and whisker plots indicate the 25[th], 50[th], and 75[th] quartiles, with the whiskers extending to 1.5 times the inner quartile range. The open dots indicate the mean values for each instrument within smoke.

consumed during western U.S. forest fires. In addition, the concentrations of pollutants generated and observed during the chamber studies were approximately an order of magnitude smaller than those observed during the prescribed grassland fires. For reference, maximum $PM_{2.5}$ concentrations observed during the prescribed fires were in excess of 50 mg m[-3] while maximum chamber $PM_{2.5}$ concentrations

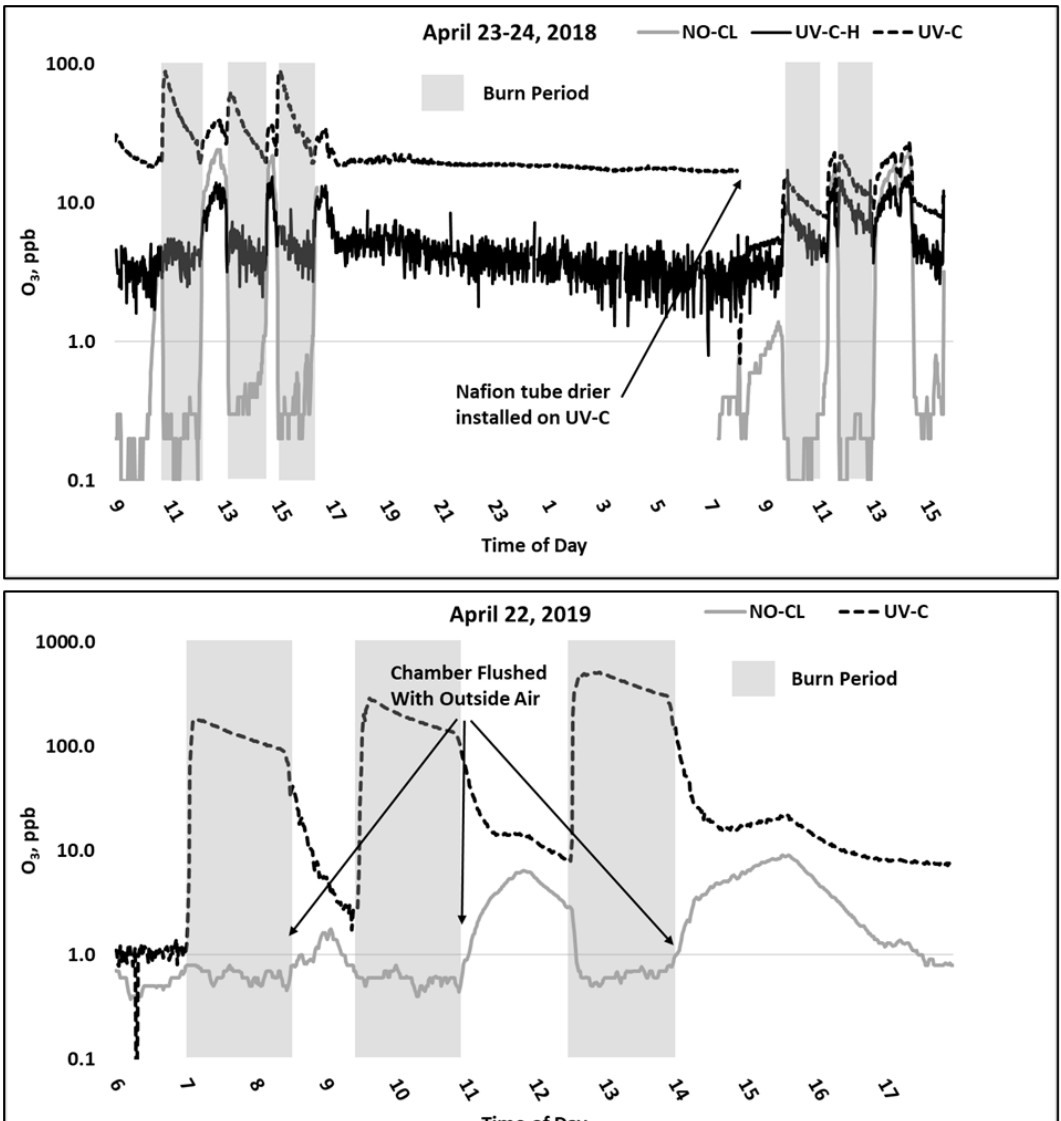

369

**Figure 4.** Time series example of USFS chamber burn $O_3$ results from the NO-CL, UV-C, and UV-C-H (2018 only) from April 23-24, 2018 (top) and April 22, 2019 (bottom). $O_3$ concentrations are displayed in a logrithmic scale. The post burn calibration checks on April 23, 2018 revealed a +8 % bias in the NO-CL method and a -2 % bias in the UV-C-H method. These biases were evident during the chamber flush periods on that day. Each analyzer was re-zeroed and spanned resulting in the elimination of the bias between the two methods as observed in the results from the subsequesn day (April 24, 2018).

were less than 2 mg m$^{-3}$ range. Regardless of these differences, there were still significant (order of magnitude or more) differences in the measurement results between the different FEM $O_3$ instruments

operated during both the 2018 and 2019 chamber studies. The NO-CL method showed identical trends to
those observed during the grassland burns in that its measured $O_3$ concentrations dropped to near zero
during the active burning periods as indicated in Fig. 4 (active burning periods shaded in grey). The only
periods when significant $O_3$ concentrations were measured by the NO-CL method was when outside air
was brought in to flush the chamber in between burns. The post burn calibration checks on April 23, 2018
revealed a +8 % bias in the NO-CL method and a -2 % bias in the UV-C-H method. These biases were
evident during the chamber flush periods on that day. Each analyzer was re-zeroed and spanned resulting
in the elimination of the bias between the two methods as observed in the results from the subsequesn day
(April 24, 2018)." No other calibration corrections werer made during the 2018 and 2019 chamber
studies. As in the grassland fire plumes, the UV-C method showed increased $O_3$ concentration (positive
analytical artifact) readings that were correlated in time with CO and $NO_2$; See Supplementary Figs. S9
and S10. Similarly, the UV-C-H instrument also showed increased positive analytical artifacts during the
chamber burns, but with absolute measurement values about an order of magnitude smaller than the UV-
C instruments. The SL-UV method gave similar results to the NO-CL method during both the 2018 and
2019 chamber burns. Newly added during the 2019 burns, the UV-G method (2B 211-G) gave mixed
results: at times it provided similar results compared to the NO-CL and SL-UV methods, and at others it
provided results in line with those provided by the UV-C method. See Supplementary Fig. S5 for the 2019
chamber burn time series plot. The burn average $O_3$ concentrations from the 2018 and 2019 chamber
burns are presented in Fig. 2.
During the 2018 chamber burns the UV-C results were biased high by 15-20 ppb even during non-burn
(i.e., overnight) periods as evident in Fig. 4 (top panel) and Fig. S4. The initial hypothesis was that the
bias was associated with high chamber backgrounds of interfering species due to years of heavy burning
in the chamber. However, it was later discovered during a subsequent summer/fall 2018 ambient air study
in North Carolina in the absence of smoke, that sampling heavy smoke plumes during the fall 2017
prescribed grassland burns followed by subsequent storage of the UV-C analyzer, irreversibly damaged
the $MnO_2$ scrubber in the UV-C instrument. It is hypothesized that the damage resulted in the scrubber
removing some of the interfering species in additon to ozone, preventing them from being subtracted off
as background in the reference measurment, and subsequent detection as ozone (positive bias) during the
measurement cycle. The effect of the bias was observed mainly when sampling ambient/chamber air and
not readily observed during routine calibration checks (zeroes and spans) except for an increase in the
time required to obtain stable zero and span values. The bias was not observed during any of the 2017
prescribed grassland burns. During the summer/fall 2018 North Carolina study and prior to the start of
the 2019 chamber burns, a new $MnO_2$ scrubber was installed and resulted in a significant and immediate
reduction of the observed high bias, shown in Fig. 4 (bottom panel) and Fig. S5.

### 412 3.3 Methodological Influence on Ozone Measurements in Biomass Burning Smoke

As discussed in Sections 3.1 and 3.2, there are large (order of magnitude level) differences in $O_3$
concentration measurement results obtained from the FRM (NO-CL) and the FEM UV photometric with
catalytic scrubber (UV-C) $O_3$ methods. The extremely low $O_3$ concentrations measured by the NO-CL
instrument is consistent with $O_3$ depletion in the presence of high $NO_x$ concentrations (up to ppm levels)
observed in the grass burning plumes and during chamber burns. The reaction between NO and $O_3$ is
rapid and occurs on the timescales of seconds to minutes. As a result, high NO in the fresh biomass
combustion plumes will efficiently titrate out $O_3$ leading to near-field depletion within the plumes relative
to background concentrations. There was no sign of a positive interference in the NO-CL monitors, and
it remains the most robust and accurate routine method for $O_3$ measurement in fresh and downwind
biomass burning plumes.

In contrast with the NO-CL FRM instrument results, the UV-C FEM results showed substantial increases
in reported $O_3$ concentrations in the fresh biomass burning plumes. There is no known pathway for direct
$O_3$ emission from biomass burning, and the proximity (meters to hundreds of meters) and timescales
(seconds to minutes travel time from the combustion source to measurement) involved are too short for
the usual $NO_x$ – VOC photochemistry to produce secondary $O_3$. Further, since the FSL chamber interior
is not exposed to sunlight, photochemistry would not have been active in the Missoula laboratory burns.
For the purposes of this work, the positive analytical artifact in the UV-C method, termed $\Delta O_{3(UV-C)}$, is
estimated using Eq. (6) as the difference between UV-C and the NO-CL $O_3$ concentration measurement
results for the same time period:

$$\Delta O_{3(UV\text{-}C)} = UV\text{-}C - NO\text{-}CL \qquad\qquad (6)$$


Figure 5 shows "in plume" regressions between $\Delta O_{3(UV\text{-}C)}$ and the FRM measurement and CO for the

three measured prescribed grassland burns in 2017 (Supplementary Fig. S6 shows the time series of


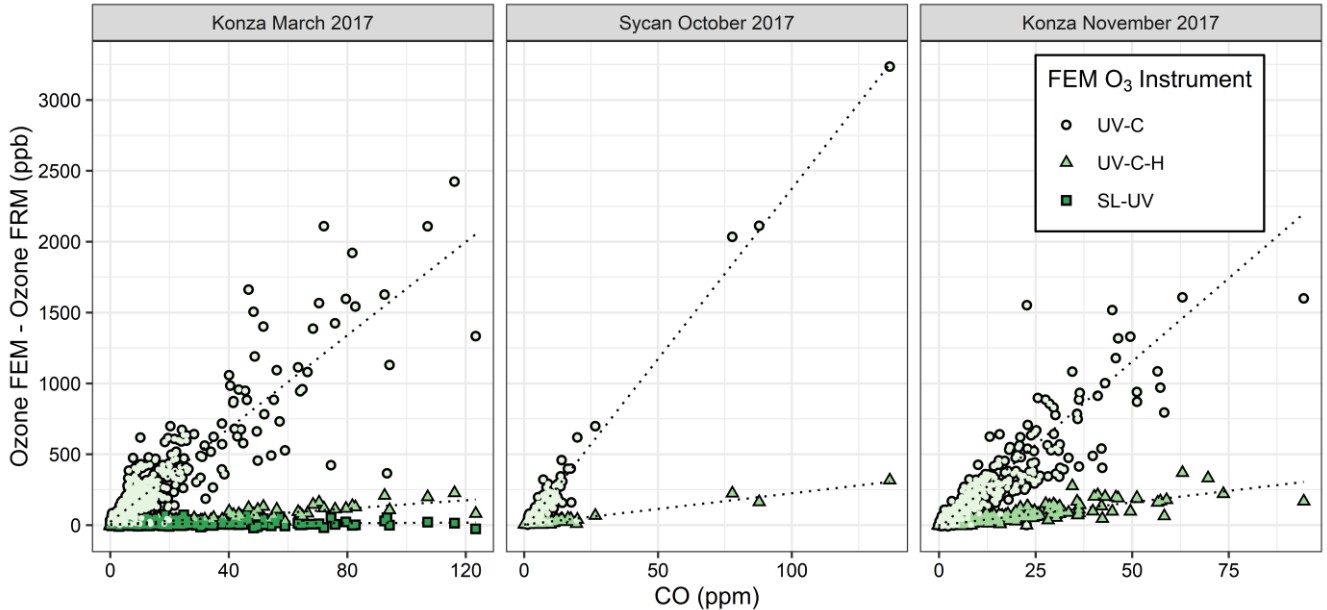


**Figure 5.** Scatter plots between FEM and FRM $O_3$ differences and the CO measurements within the
grassland fires smoke plumes. The FEM measurements are differentiated by color and shape. The SL-UV
method was only run during the Konza March 2017 measurements.

$\Delta O_{3(UV\text{-}C)}$ and CO). Figure 5 and Supplementary Fig. S6 show good correlations within the smoke plumes.

The average and maximum values of $\Delta O_{3(UV\text{-}C)}$ are summarized in Table 3. It is hypothesized that the

large "$O_3$" measurement observed in the UV-C method results from a positive interference or artifact,

likely linked to VOC emissions in the grassland burn plumes. VOCs are emitted in higher concentrations

from the smoldering phase of combustion, which is also characterized by large CO emissions (Yokelson

et al., 1996; Yokelson et al., 1997), so a correlation between CO and $O_3$ artifact would support the

hypothesis of a VOC-linked interference for the UV-C instruments. This is also consistent with observed

450

**Table 3: Ozone artifact ($\Delta O_3$) averages, maximum values, and CO, NO$_2$, and THC averages from the prescribed fire and USFS chamber burns as measeured by the UV-C, UV-C-H, and UV-G instruments.**

| Study | $\Delta O_3$ avg. (ppb) | $\Delta O_3$ max (ppb) | CO avg. (ppm) | NO$_2$ avg. (ppb) | THC avg. (ppm) |
|---|---|---|---|---|---|
| **$\Delta O_{3(UV-C)}$** | | | | | |
| Mar. 2017 Konza Prairie (KS) | 295.8 | 2423.7 | 15.8 | 147.3 | - |
| Oct. 2017 Sycan Marsh (OR) | 170.2 | 3235.5 | 8.5 | 60.5 | 2.7 |
| Nov. 2017 Konza & Tallgrass Prairies (KS) | 330.0 | 3156 | 14.1 | 156.9 | 4.0 |
| Apr. 2018 USFS Chamber (MT) | 36.5 | 309.6 | 3.8 | 35.6 | 2.8 |
| Apr. 2019 USFS Chamber (MT) | 66.9 | 530.9 | 2.1 | 31.7 | 4.8 |
| **$\Delta O_{3(UV-C-H)}$** | | | | | |
| Mar. 2017 Konza Prairie (KS) | 42.8 | 227.1 | 15.8 | 147.3 | - |
| Oct. 2017 Sycan Marsh (OR) | 21.1 | 316.4 | 8.5 | 60.5 | 2.7 |
| Nov. 2017 Konza & Tallgrass Prairies (KS) | 40.2 | 369.0 | 14.1 | 156.9 | 4.0 |
| Apr. 2018 USFS Chamber (MT) | 7.2 | 136.8 | 3.8 | 35.6 | 2.8 |
| **$\Delta O_{3(UV-G)}$** | | | | | |
| Apr. 2019 USFS Chamber (MT) | 22.9 | 376.8 | 2.1 | 31.7 | 4.8 |
| **$\Delta O_{3(SL-UV)}$** | | | | | |
| Mar. 2017 Konza Prairie (KS) | 8.3 | 74.2 | 15.8 | 147.3 | - |
| Apr. 2018 USFS Chamber (MT) | 0.5 | 11.5 | 3.8 | 35.6 | 2.8 |
| Apr. 2019 USFS Chamber (MT) | 1.7 | 32.1 | 2.1 | 31.7 | 4.8 |

VOC interferences in previous studies (Grosjean and Harrison, 1985; Kleindienst et al., 1993; Spicer et al., 2010) and observed following fireworks (Fiedrich et al., 2017; Xu et al., 2018).

The presence of a Nafion®-based humidity conditioning system (Nafion® tube dryer) significantly reduced the magnitude of the observed artifact as evident by comparing the UV-C and UV-C-H results shown in Figs. 1-3 and Supplementary Figs. S1 – S5. As with the UV-C method, the artifact in the UV-C-H method, $\Delta O_{3(UV-C-H)}$, is calculated using Eq. (7) as the difference between UV-C-H and the NO-CL O$_3$ concentration measurement results for the same time period:

$$\Delta O_{3(UV-C-H)}=UV\text{-}C\text{-}H – NO\text{-}CL \qquad (7)$$

The addition of the Nafion®-based humidity conditioning system reduces the magnitude of the $\Delta O_{3(UV-C-H)}$
artifact by approximately an order of magnitude as compared with the UV-C method. This is further
illustrated in the 2018 chamber burns, where prior to beginning the final burn day on April 24, 2018, a
Nafion® tube dryer (PermaPure, MD Monotube Dryer Series) was installed on the UV-C method (Thermo
49i) in effect, converting it to a UV-C-H method. As shown in Fig. 4 and Supplementary Fig. S4, the
additon of the Nafion® tube dryer significantly reduced the $\Delta O_{3(UV-C)}$ artifact to a point comparable with
that observed in the UV-C-H method (2B 205). A possible explanation for this effect is presented and
discussed in Section 3.5. In addition, the previously described bias related to the damaged $MnO_2$ scrubber
was also reduced upon addition of the Nafion® dryer to the UV-C method.

For the March 2017 Konza Prairie study (Fig. 1) and the 2018 and 2019 USFS chamber studies (Fig. 3)
the SL-UV instrument concentration results were comparable to, although noisier and slightly higher than,
the NO-CL reference instrument. On numerous occasions during the prescribed and chamber burns, the
SL-UV instrument shows short (i.e. one-minute data point) positive or negative excursions that are not
also observed in the NO-CL results. In addition, these excursions are not correlated with changes in CO
concentrations. Because the SL-UV is a dual cell instrument that measures $O_3$ by comparing the
absorbance of two cells, it is critical in highly dynamic environments (such as during this study) that both
cells be measuring the same air at the same time. A slight difference in flow rates or residence times
between the two pathways (or a delay in one pathway relative to the other) will cause short term variability
in the difference between the two cells. Although this does not pose an issue for longer time averaging
(i.e. hourly data) under stable conditions, the dynamic nature of biomass burning plumes (i.e. changing
on the order of seconds) and short time averages (i.e. minute) can create issues (noise) for the SL-UV
method.

Significant analytical artifacts were observed for FEM UV photometric $O_3$ instruments with (UV-C-H)
and without (UV-C) Nafion®-based humidity conditioning system, where it appears that the dual effect
of ambient humidity fluctuations and VOC interferences caused large positive over-measurement of "in-
smoke" $O_3$ concentrations. Chemiluminescence monitors are highly specific to $O_3$ and have long been
known to be free of VOC interferences (Long et al., 2014; U.S. EPA, 2015). However, studies have shown
that the chemiluminescence method can be impacted by changes in relative humidity (Kleindienst et al.,
1993). As such, upon promulgation in 2015, the new NO-CL FRM regulatory text requires a humidity
correction/dryer system to eliminate the potential water vapor interference. As configured from the
manufacturer, the NO-CL based Teledyne-API Model T265 instrument operated during this comparative
study employs Nafion$^®$ drying technologies to reduce or eliminate the water vapor interferences. The use
of a chemical (NO) scrubber for UV photometric instruments (such as the 2B Technologies Model 211)
is very specific to $O_3$ and shows a much better response than the catalytic scrubber instruments,
performing almost as well as the NO-CL FRM, and has significant potential as a low-interference $O_3$
method. Of the catalytic scrubber photometric instruments those with Nafion$^®$-based humidity
equilibration (2B Technologies Model 205) perform significantly better than those without (Thermo 49
series).

In areas highly impacted by smoke or for studies focusing on biomass burning plumes, the use of a NO-
CL FRM instrument is highly recommended as it was found to be essentially interference-free. These
instruments are anchored to absolute $O_3$ concentrations through the use of certified $O_3$ calibration sources,
many of which are based on UV photometry. The newest generation of commercially-available NO-CL
FRM instruments, including that used here (the Teledyne T265), have a built-in drying system to correct
for the humidity artifacts that affected earlier generation chemiluminescence instruments (Kleindienst et
al., 1993), making remaining interferences negligible compared to other technologies.

The gas-phase chemical scrubber UV instrument (2B 211), did not perform as well as the FRM under the
prescribed grassland burns or chamber experimental conditions tested here, with the high time resolution
(1-minute) data showing a much higher degree of variability than the NO-CL FRM instrument. We
hypothesize that the main factor driving this divergence between this method and the NO-CL FRM is the
dual-cell differential configuration of the instrument, which is not conducive to rapidly changing
concentrations in $O_3$ or other absorbing gases, such as VOCs.

In smoke-impacted monitoring situations where the use of a UV photometric instrument is still preferred or required, the choice of a monitor with humidity equilibration provides a significant analytical improvement over those monitors without the humidity corrections. In the absence of an instrument with a Nafion® tube dryer and in non-regulatory applications, a dryer can be installed before the inlet or measurement cells to reduce the interference as was demonstrated on the final day of the 2018 Missoula chamber burns. This will have the added benefit of reducing positive biases from humidity and reducing equilibration time for calibrations (especially when switching from high humidity ambient air to dry calibration gases).

### 3.4 Magnitude of Ozone Artifact in Fresh Biomass Burning Plumes Relative to Markers of Combustion

It is difficult to estimate an absolute magnitude or correct for the observed $O_3$ analytical artifact since primary emissions from biomass combustion are highly variable and depend upon the fuel loading, fuel type and condition, phase of the fire, and the burn conditions (Yokelson et al., 1996; Yokelson et al., 1997). However, assuming the interference is driven primarily by VOCs, the artifact should be correlated with the excess CO ($\Delta CO = CO_{plume} - CO_{background}$). Because $CO_{background}$ during the prescribed grassland burns was below 200 ppb (relative to typical conditions of >2 ppm in the plume), $\Delta CO$ is estimated as the total measured CO concentration. A simplified view of biomass combustion assumes an approximate linear combination of two dominant emission phases, flaming combustion (characterized by emission of highly oxidized compounds, such as $CO_2$, $NO_x$, and $SO_2$), and smoldering combustion (characterized by emission of reduced or mixed oxidation state compounds, such as CO, $CH_4$, $NH_3$, $H_2S$, and most VOCs) (Yokelson et al., 1996; Yokelson et al., 1997). Because the majority of VOCs are in a reduced or mixed oxidation state, they tend to be co-emitting with CO during smoldering combustion, and the VOC concentrations tend to be highly correlated with CO in fresh biomass burning plumes (Yokelson et al., 1996). Scatterplots comparing the FEM instrument artifacts ($\Delta O_{3(UV-C)}$) and CO for the three prescribed grassland burning periods are shown in Fig. 5. Regression statistics of the comparison of $\Delta O_{3(UV-C)}$ and $\Delta O_{3(UV-C-H)}$ with CO and THC for grassland burns are given in Table 4. The magnitude of the artifact (estimated by the slope of the regression line of the CO vs $\Delta O_3$ comparison), in ppb apparent $O_3$ per ppm

CO, ranges between 16 - 24 ppb ppm$^{-1}$ for the UV-C instrument, and 1.5-3 ppb ppm$^{-1}$ for the instrument
with humidity correction (UV-C-H). It is important to point out that CO, in and of itself, is not considered
to be an interfering species in the UV photometric determination of $O_3$ in that CO absorbs in the infrared
(IR). The slight differences in the magnitude of the artifacts (fitted regression slopes) along with the low
uncertainty (standard errors) values indicate that the magnitude of the artifact may be influenced by local
conditions that make each burn unique. Such condiitons might include meteorological conditions, fuel
composition, fuel moisture content, and times spent in combustion phase (flaming vs smoldering). Similar
to CO, THCs and $NO_2$ are indicative of combustion processes and are correlated with $\Delta O_3$ as given in
Table 4 and Supplementary Figs. S7 and S8. In terms of THC, the magnitude of the artifact, in ppb
apparent $O_3$ per ppm THC, is significantly higher at ~88 ppb ppm$^{-1}$ for the UV-C instrument and ~13 ppb
ppm$^{-1}$ for the UV-C-H instrument. Both the prescribed grassland and Missoula chamber burns resulted in
what would be considered high PM concentrations (2-50 mg m$^{-3}$). These high PM concenttrations
however, are not considered to be interfering due to the presence of the inline particle filter assemblies
described in Sections 2.2 and 2.6.

Since the CO concentrations (from upwind fires) observed at most stationary sites from fire plumes are
usually on the order of one ppm to greater than 10 ppm (Landis et al., 2018), it is reasonable to assume
that $O_3$ artifacts in the range of 15 ppb to greater than 250 ppb can be observed when employing a UV-C
method. Similarly, $O_3$ artifacts in the range of 1.5 to above 30 ppb might be observed at smoke-impacted
sites monitoring with UV-C-H methods. As such, Nafion®-based humidity conditioning systems are
highly recommended for use if employing UV photometric methodology for monitoring $O_3$ in areas
impacted by wildfires or prescribed burns. As stated previously and as seen in Fig. 3 and Table 3, $O_3$
artifacts were observed during the Missoula chamber 2018 and 2019 burns in both the UV-C and UV-C-
H methods, although reduced compared to the prescribed grassland burns. The presence and magnitude
of the $O_3$ artifact strongly suggests that smoke generated from fuels typical of forests in the western United
States also result in a measurement interference in UV photometric methods. Since downwind $O_3$
production in biomass burning plumes is a significant issue in fire impacted regions, having reliable,
interference-free methods is critical for assessing the contribution of wildland fires to ambient $O_3$ levels.

**Table 4: Regression statistics for the ozone artifact ($\Delta O_3$) versus CO and THC for UV photometric instruments without (UV-C) and with (UV-C-H) a Nafion®-based humidity equilibration system during the 2017 prescribed grassland burns.**

| Study | Slope (ppb/ppm) | Intercept (ppb) | $r^2$ | n |
|---|---|---|---|---|
| **$\Delta O_{3(UV-C)}$ vs CO** | | | | |
| Mar. 2017 Konza Prairie (KS) | 16.46($\pm$0.34)[a] | 18.53($\pm$6.72)[b] | 0.79 | 653 |
| Oct. 2017 Sycan Marsh (OR) | 24.02($\pm$0.25) | -28.05($\pm$2.73) | 0.96 | 295 |
| Nov. 2017 Konza & Tallgrass Prairies (KS) | 23.51($\pm$0.73) | -20.8($\pm$13.03) | 0.74 | 461 |
| **$\Delta O_{3(UV-C)}$ vs THC** | | | | |
| Nov. 2017 Konza & Tallgrass Prairies (KS) | 87.14($\pm$3.74) | -85.36($\pm$18.63) | 0.59 | 461 |
| **$\Delta O_{3(UV-C-H)}$ vs CO** | | | | |
| Mar. 2017 Konza Prairie (KS) | 1.46($\pm$0.04) | 0.87($\pm$1.03) | 0.80 | 163 |
| Oct. 2017 Sycan Marsh (OR) | 2.21($\pm$0.05) | 3.44($\pm$0.54) | 0.88 | 296 |
| Nov. 2017 Konza & Tallgrass Prairies (KS) | 3.24($\pm$0.09) | -1.17($\pm$1.67) | 0.77 | 461 |
| **$\Delta O_{3(UV-C-H)}$ vs THC** | | | | |
| Nov. 2017 Konza & Tallgrass Prairies (KS) | 13.27($\pm$0.39) | -14.53($\pm$1.92) | 0.75 | 461 |
| **THC vs CO** | | | | |
| Nov. 2017 Konza & Tallgrass Prairies (KS) | 0.21($\pm$0.004) | 1.55($\pm$0.08) | 0.79 | 461 |

[a]Standard error or uncertainty of the linear regression slope in ppb/ppm
[b]Standard error or uncertainty of the linear regression intercept in ppb

Figure 6 gives a detailed time series view of $\Delta O_{3(UV-C)}$ and CO from two burn days from 2018 and a single day during 2019. As indicated, $\Delta O_{3(UV-C)}$ and CO appear to be correlated in time but when performing linear regression comparisons of $\Delta O_{3(UV-C)}$ and CO during each years chamber burns as a whole, correlations tend to be poor. We suspect the positive $O_3$ bias is driven by one or more VOCs (likely oxygenated VOCs). In fresh smoke the excess concentrations of individual VOCs ($\Delta X$), and VOC sums ($\Delta VOC$), tend to be highly correlated with $\Delta CO$ (Yokelson et al., 1999; Gilman et al. 2015). The emission ratios of individual VOCs to CO ($\Delta X/\Delta CO$) can vary considerably with combustion conditions such as fuel type and condition (e.g. moisture content and decay state), fuel bed properties, such as bulk density, and the relative mix of flaming and smoldering combustion (Gilman et al. 2015; Koss et al., 2017). Additionally, the response of $\Delta X/\Delta CO$ to burn conditions varies among VOCs. When each burn is considered individually or in groups with similar conditions, the correlations between $\Delta O_3$, CO, and THC are enhanced. An example of this behavior is shown in Supplementary Fig. S10. For the chamber burns


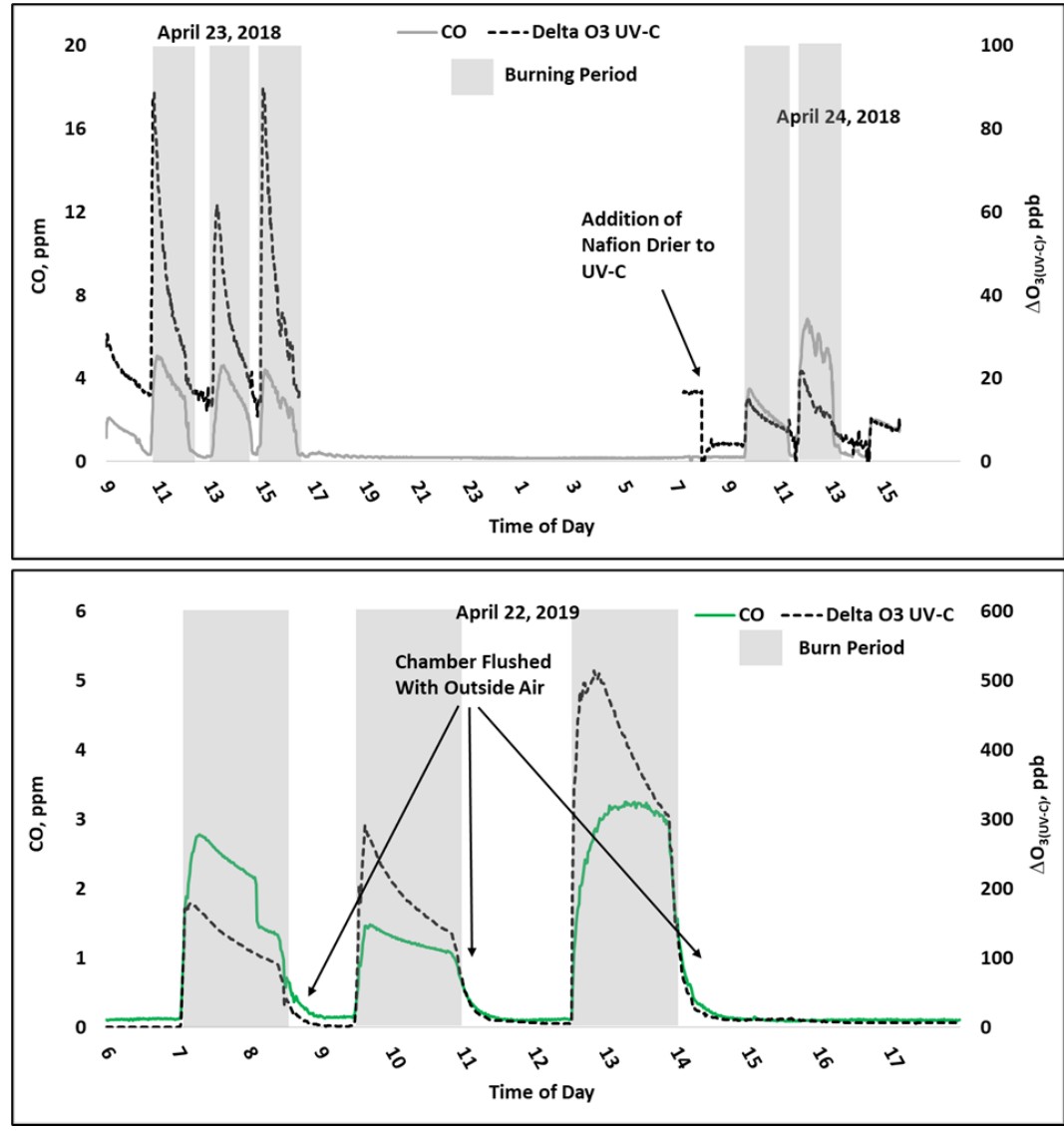


**Figure 6.** Time series example of USFS chamber burn $\Delta O_3$(UV-C) and CO concentration results from April 23-24, 2018 (top) and April 22, 2019 (bottom).


the magnitude of the ozone artifacts in ppb apparent $O_3$ per ppm CO, ranges between 6 - 210 ppb ppm[-1] for the individual burns. $R^2$ and standard error values were consistent with those observed dring the prescribed burns (see Table 4). The lack of a consistent relationship between the $O_3$ artifact and $\Delta$CO

across all FSL chamber burns, while observing a good correlation for individual burns, likely reflects the
variable response of artifact producing emission(s) to the different combustion conditions of the burns.

One interesting observation from the data obtained from both the prescribed grassland and chamber burns
is the order of magnitude difference in the average and maximum $O_3$ artifact between the UV-C and the
UV-C-H instruments as shown in Table 3. Considering that the prescribed grassland and chamber burns
were conducted under dry (RH < 50%) conditions, the size of the difference (as large as hundreds of ppb)
cannot be explained purely by the previously observed relative humidity effects on measurements (Leston
et al., 2005; Wilson et al., 2006), suggesting that the Nafion® dryer is directly impacting the concentrations
of other interferents in the sample stream.

### 3.5 Potential Reason for Lower Artifacts with Methods Employing Nafion®-based Humidity Equilibration

Nafion® is a sulfonated tetrafluoroethylene polymer that is highly permeable to water but shows little
permeability to many other organic and inorganic species (Mauritz et al., 2004). As a result, Nafion®-
based drying systems are often used as part of sample preparation or conditioning systems in analytical
chemistry to remove water vapor from sample streams prior to sample analysis. Nafion® membranes were
introduced to some $O_3$ monitors as a method to address humidity effects observed in UV-C $O_3$ monitors,
particularly when there are rapid changes in relative humidity level (Wilson and Birks, 2006). Humidity
can affect the transmission of the UV light through the detection cell and catalytic $O_3$ scrubbers can
modulate the water vapor in the scrubbed channel by acting as a temporary reservoir, resulting in
significant positive or negative $O_3$ interferences during rapid swings in relative humidity Wilson et al.,
2006). Adding a Nafion®-based equilibration dryer immediately prior to the measurement cells reduces
this water vapor interference without affecting $O_3$ concentrations, and thus significantly reduces the
humidity artifacts in UV photometric $O_3$ instruments.

Despite the high selectivity of Nafion® to water vapor, it does demonstrate partial to complete
permeability to various VOC or semivolatile organic compounds. Nafion® membranes are highly
permeable to alcohols, amines, ketones, and some water-soluble ethers (Baker, 1974), as well as some
biogenic oxygenated compounds (Burns et al., 1983). In addition, Nafion® membranes have been shown
to catalyze the decomposition and rearrangement of monoterpene compounds (Burns et al., 1983).
Systematic study of Nafion® permeability and reactivity for polar and oxygenated compounds has been
limited, with most users of Nafion® membranes basing their use on operational testing and confirmation
for the targeted use.

The significant (order of magnitude) reduction in the $O_3$ artifact with the addition of a Nafion®-based
dryer to the UV-C suggests that the Nafion® dryer is directly impacting the major interfering species
which was hypothesized to be VOCs emitted during combustion processes. The species that are
responsible for most of the $O_3$ artifact in UV-C $O_3$ instruments would have to be permeable through
Nafion® membranes or reactive with Nafion® membranes, be scrubbed by solid-phase, catalytic $O_3$
scrubbers (such as $MnO_2$ or hopcalite), and would have a significant absorption cross section around 254
nm. The absorption cross-section of $O_3$ around 254 nm is on the order of $10^{-17}$ $cm^2$ molecule$^{-1}$ (Molina
and Molina, 1986), which means species with absorptions around $10^{-17}$ $cm^2$ molecule$^{-1}$ at 254 nm would
be potential interfering species. As a class, aromatic VOCs and specifically oxygenated aromatic species
(and other polar derivatized species) absorb strongly in this region of the UV spectrum, and their potential
permeability through Nafion® membranes result in them being likely compounds to interfere in UV-C
instruments. As an example, aromatic aldehydes such as o-tolualdehyde and p-tolualdehyde absorbances
around $5x10^{-18}$ $cm^2$ molecule$^{-1}$ and $4x10^{-18}$ $cm^2$ molecule$^{-1}$, respectively (Etzkorn et al., 1999). Both 2,4-
dimethylbenzaldehyde and 2,6-dimethylbenzaldehyde have absorption cross sections above $10^{-17}$ $cm^2$
molecule$^{-1}$ at 254 nm (El Dib et al., 2008). Baker (1974) found 75% of benzaldehyde was removed by a
Nafion® membrane, meaning that the Nafion® permeability of tolualdehydes and dimethylbenzaldehydes
is also likely to be high. In addition, benzaldehyde was almost quantitatively removed by several
commercial catalytic $O_3$ scrubbers, including the Thermo 49i $MnO_2$ catalytic scrubber (Kleindienst et al.,
1993), so similar aldehydes are likely to behave in a similar manner. Therefore, substituted aromatic
aldehyde species are one class of compounds that fit the necessary criteria for causing the interference on
the UV-C while having a reduced interference on the UV-C-H instrument. Future work examining the
potential interferences from different species (or classes of species) on a species or class specific basis
are required to confirm this potential mechanism and suggest others.
**4 Implications**
Wildland fires (wildfires and prescribed fires) emit significant amounts of VOCs and NOx, two important
precursors in the photochemical formation of tropospheric $O_3$. Therefore, it is not surprising that large
increases in $O_3$ are routinely reported at ambient monitoring sites downwind from wildland fires (DeBell
et al., 2004; Bytnerowicz et al., 2010; Preisler et al., 2010; Jaffe et al., 2012; Bytnerowicz et al., 2013;
Jaffe et al., 2013; Lu et al., 2016; Lindaas et al., 2017; Baylon et al., 2018; Liu et al 2018; McClure and
Jaffe, 2018). For example, Buysse et al. (2019) examined regulatory air monitoring data from 18 cities
over a five period, and found that July – September exceedances of NAAQS for $O_3$ were far more common
on days with known wildland fire smoke impacts (4.6%) than those without (<0.1%). However, the results
of this study suggest caution when interpreting UV photometric method $O_3$ measurements under
conditions of wildfire smoke impact due to the significant positive artifacts that were observed. The
analytical artifacts were also shown to be positively correlated with widely used markers of combustion
such as CO and THC suggesting that the artifacts arise from photometric measurement interferences by
VOCs and further complicating the interpretation of smoke impacted UV photometric $O_3$ data. As
described in section 3.4, it reasonable to assume that $O_3$ artifacts in the range of a few ppb to greater than
250 ppb in addition to actual photochemically formed $O_3$ can be observed when employing UV
photometric methods at sites downwind from fires.

A detailed example of observed artifacts in the UV photometric method occurred during the 2016 Fort
McMurray Horse River wildfire in Alberta, Canada, where elevated "$O_3$" concentrations were observed
at multiple community based air monitoring sites utilizing UV-C instruments in the vicinity of the fire
(Landis et al., 2018). Reported "$O_3$" concentrations reached maximum hourly concentrations in excess of
1500 ppb using UV-C methods at night (between 10:00 PM and 5:00 AM local) in the absence of
photochemistry and were positively correlated with the combustion markers NO and non-methane
hydrocarbon (NMHC). Peaks in $O_3$ concentration are expected to be negatively correlated with peaks in
NO concentration as it rapidly titrates $O_3$ to $NO_2$, and the authors hypothesized that UV photometric
measurement artifacts may have been responsible for the unexpected observations.

The findings from this research effort and the observations from ambient studies (Landis et al., 2018)
raise concerns that routine regulatory monitoring and wildland fire research study $O_3$ measurements
utilizing UV photometric FEM instruments may be reporting positive measurement artifacts as $O_3$ during
smoke impacted events. Some studies have hypothesized that rapid photochemical processing was
responsible for reported elevated $O_3$ concentrations reported in downwind wildfire plumes (Liu et al.,
2017). Since downwind $O_3$ production in biomass burning plumes is a significant issue in fire impacted
regions, having reliable, interference-free methods is critical for assessing the contribution of wildland
fires to ambient $O_3$ levels and developing/validating accurate deterministic air quality models. Air quality
researchers and environmental regulators are strongly encouraged to utilize NO-CL FRM $O_3$ instruments
in areas routinely impacted by wildland fire smoke.
**5 Conclusions**
In this study, we compare two different $O_3$ measurement methods (chemiluminescence and UV
photometry) in fresh biomass burning plumes from prescribed grassland fires and during controlled
chamber burns. Within the UV photometry category, we look at two different technologies, one using a
gas-phase chemical scrubber (NO) and the second using solid phase catalysts to scrub $O_3$ from analytical
reference channels. Among the UV photometric instruments employing solid phase catalytic scrubbers,
we evaluated and compared methods that include a Nafion®-based humidity equilibration system with
those that do not.

The NO-CL method, recently promulgated as the $O_3$ FRM, performed well, even in fresh plumes, whereas
the UV photometric methods displayed varying degrees of positive measurement artifacts. The UV
photometric method employing the dynamic NO gas phase scrubber performed comparably with the NO-
CL method but was not well suited to the rapidly varying concentrations of VOCs in the smoke plumes.
The catalytic scrubber photometric methods demonstrated positive analytical artifacts that were correlated
with CO and THC concentrations (both biomass burning plume indicators). There was a significant
difference between the catalytic scrubber UV instruments with and without Nafion®-based humidity
correction, with the dryer system reducing the positive $O_3$ artifact by an order of magnitude as compared
with the UV photometric method employing no humidity correction. The observed reduction in artifacts
cannot be attributed only to elimination of the relative humidity/water vapor interferences and likely result
from post-scrubber equilibration or reaction of Nafion®-permeable VOCs prior to the measurement cell.
The results of this study strongly suggest that careful consideration be given to employed measurement
methods when monitoring $O_3$ concentrations in regions where impacts from biomass burning routinely
occur due to the significant impact of potential measurement interferences. In addition to consideration
of operating methods containing Nafion®-based humidity condition systems, attention should be focused
on the scrubbers employed by UV photometric methods and the adverse effects that operation in smoke
may have on those scrubbers. Further research is being conducted to evaluate the magnitude of the artifact
in the UV photometric method at routine monitoring sites that are often impacted by wildland fire smoke
events under the EPA Mobile Ambient Smoke Investigation Capability (MASIC) program (U.S. EPA
726 2019).

**Data Availability**
Datasets related to this manuscript can be found at https://catalog.data.gov/dataset/epa-sciencehub.

**Author Contributions**
Russell W. Long served as principal investigator and prepared the manuscript with contributions from all
co-authors.  Russell W. Long, Andrew Whitehill, Andrew Habel, Maribel Colón, Shawn Urbanski, and
Matthew S. Landis performed the prescribed grassland fire and FSL chamber-based data collection and/or
analysis.  Surender Kaushik performed supervisory review of this research effort and corresponding
manuscript.

**Competing Interests**

The authors declare that they have no conflict of interest.

**Disclaimer**

The views expressed in this paper are those of the authors and do not necessarily reflect the views or policies of EPA. It has been subjected to Agency review and approved for publication. Mention of trade names or commercial products do not constitute an endorsement or recommendation for use.

**Acknowledgements**

The EPA through its Office of Research and Development (ORD) funded and conducted this research. We thank Kansas State University, The Nature Conservancy, Konza Prairie Biological Station staff, Sycan March Preserve staff, Tallgrass Prairie National Preserve staff, numerous burn crews, Brian Gullet (EPA), Cortina Johnson (EPA), Melinda Beaver (EPA), Libby Nessley (EPA), and Kyle Digby (Jacobs).

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
