# Peer review of "Comparison of Ozone Measurement Methods in Biomass Burning Smoke: An evaluation under"

_Atmospheric Measurement Techniques, 2020_

## Referee Comment (RC1) · Anonymous Referee #1 · 7 Oct 2020

Review of Long et al., manuscript ID amt-2020-383, "Comparison of ozone measurement methods in biomass burning smoke: an evaluation under field and laboratory conditions"

General comments.

This paper uses measurements in highly concentrated fire plumes (within 100m of wildland grass fires, and in controlled burns at the Missoula Fire Lab) to assess interferences in UV absorption measurements of ozone at 254 nm.

This paper is motivated by the ozone measurements of UV absorption instruments, and the health impacts from that ozone. Large increases in ozone may be observed after

precursor NOx and VOC have had time to react. The time scale to produce that ozone is highly dependent on plume dilution, which itself is highly variable in time, but typically takes place over hours since emission. A fundamental question: for direct emissions, the interfering species will also be diluted, such that the lowest interferences may be expected at the highest levels of plume ozone. Secondary production of UV-active hydrocarbons, e.g., production of nitroaromatics following oxidation in the presence of NO2, may dominate the ozone interference downwind. What balance of directly emitted vs. secondary species are conjectured to lead to interferences in ambient ozone measurements? Regardless of the source of the interference (primary vs. secondary), given the lack of consistency from fire to fire (or even between different implementations of the UV absorption technique) in the level of interferences measured, can the authors say what level of "fire impact" causes a non-negligible interference? Is 1 ppm of ozone acceptable? 10 ppm of ozone? Regardless, despite the experimental detail in this paper, it is not clear what ozone monitoring locations are expected to suffer from significant interferences as a result of wildfires or prescribed burns. Lacking these considerations, the paper's conclusions are qualitative at best, and by implication condemn a much larger portion of the U.S. ozone monitoring network during the fire season than I suspect is warranted. For a given UV absorption monitor, can they recommend what data to retain, and what data to eliminate because of fire impacts? Some additional clarity in the real-world effects of fire smoke on ozone monitoring is needed for this to make a novel and useful contribution to the literature.

The paper is overly long and can be shortened by removing extraneous details, repetitive text, and tables that do not provide any usefully generalizable data as suggested below. Earlier literature is not well cited and additional references are also suggested below.

Specific comments.

line 14: "... large increases in ozone are also observed downwind ..." Is this always true?

line 32 (and lines 38 and 182 and elsewhere): The NO-induced chemiluminescence measurement of ozone is repeatedly described as "interference-free", which is misleading - it has a known dependency on water vapor, which can lead to sensitivity variations of up to 8% if not accounted for. Please rephrase.

line 52: for clarity please change to "... generates nitrogen dioxide in an electronically excited state..." The original citation is Clough and Thrush, 1966, Chemical Communications, 728, pp. 783-784.

line 93: please remove $CO_2$, as its absorption is negligible at 254 nm.

line 142: "...a supply of NO gas..." is not always needed - line 213 refers to one implementation of the "scrubberless" UV absorption method uses a supply of $N_2O$ gas, and produces NO by photolysis.

lines 230 - 237: Details of power, generator, charger, and batteries are tangential to the performance of the analyzers and could be eliminated to shorten the text.

lines 267-8: "...calibrations for THC were performed using... a methane/propane gas cylinder..." This work eventually concludes that VOCs are "likely to interfere with UV absorption measurements of O3"; no surprise there. What is surprising is the rudimentary approach to quantifying those VOCs in this manuscript. FID response factors vary with carbon number (for example, by up to a factor of 3 between methane and propane!), between aliphatic, aromatic, and cyclic structures, and with heteroatomic functionality. A sentence noting the uncertainty introduced in their measurement of VOCs (here called THC) by using only methane and propane to determine FID sensitivity would be appropriate here.

Figure 2: This is not a good graphic. There is absolutely no information conveyed by the third dimension of this graph; please turn this into a 2D bar graph and improve the legibility of the different hatches. The high level of interference from the UV-C and UV-C-H techniques overwhelms any useful information on the other techniques - suggest

[Figure]

plotting only to 50 ppb and annotating the UV-C maxima with text. These data are presented as O3 in ppb - what is the correct, or expectation value? The NO-CL data are lost in this presentation and should be emphasized as the correct value.

Figure 4: The NO-CL reference trace in the upper figure is the hardest to see; these figures coudl use some work for legibility. The text refers to positive artifacts for the UV methods during burning periods, ascribed to interferences from VOCs and PM2.5. Another problematic feature is the negative artifact when the chamber is flushed with outside air, where the UV-C method falls below the NO-CL method (bottom panel). Why is that? Did I miss the explanation?

Lines 378-388: I could not follow the confusing thread discussing how and when the MnO2 scrubber failed in these experiments - for clarity I'd recommend deleting this section and removing all data taken with an inoperative scrubber.

Table 3: Since there appears to be very large fire-to-fire and technique-to-technique variability in the interferences, with no consistent dependence on any of the variables measured, quantifying their precise values in a table seems not very useful. I'm not sure what information this table provides; what quantitative use is it? Recommend deleting.

line 498: This section recommends using Nafion dryers to minimize smoke interferences in UV absorption ozone measurements. This begs the question - under what range of conditions does the use of a Nafion dryer allow EPA to actually accept an ozone measurement by the UV absorption measurement? Please discuss.

Table 4: Same comment as for Table 3, above: "Since there appears to be very large fire-to-fire and technique-to-technique variability in the interferences, with no consistent dependence on any of the variables measured, quantifying their precise values in a table seems not very useful. I'm not sure what information this table provides; what quantitative use is it? Recommend deleting."

line 581: I would suggest the authors review and cite the use of perfluorosulfonate membrane tubing to remove UV-active hydrocarbons, e.g., in SO2 pulsed fluorescence instruments (Luke, W., 1997, JGR, 102, 16,255-16,265).

---

## Referee Comment (RC2) · Anonymous Referee #2 · 27 Oct 2020

General Comments: This study compares O3 measurement techniques in fresh, concentrate smoke plumes. The authors sample smoke plumes from both prescribed prairie grass burns and controlled chamber burns using a NO chemiluminescence measurement as the interference-free standard with which to compare several iterations of UV absorption-based measurements. This study is motivated by the prevalence of UV-based O3 analyzers at EPA air quality monitoring stations and the increasing impact of fire emissions on local and regional air quality. Although these comparisons provide insight into the potential for UV-active VOCs in smoke plumes to generate positive artifacts in the UV-based O3 measurements, a more quantitative assessment is limited by the lack of detailed VOC measurements and the inability to quantitatively

disentangle the various CO-ΔO3 regimes. The authors also suggest the role of Nafion in mitigating potential artifacts, but do not provide enough information on the relative humidity conditions during the various sampling periods or the potential for interactions between water vapor and VOC. Further, the analysis emphasizes the effects of VOC interreferences in near-fire smoke plumes but does not provide much discussion on how the potential for interference diminishes with plume age and dispersion. For example, how quickly do VOC react/diffuse to the point where their levels are no longer of concern? How many ozone monitoring sites would be practically affected by these interferences?

Specific Comments: L243-244: Is there any dependence of the artifact magnitude on distance from the active fire line? How quickly do the VOC react/diffuse to the point where their levels are no longer detectable as a positive artifact? All the measurements presented are taken within ~100 m from the fires, but any data collected from aged smoke would be a useful counterpoint.

L262: The authors mention a +/- 10% performance objective between analyzers. Do the calibrations reveal any systematic offset between the CL and UV analyzers?

In describing the prescribed and chamber burns, the authors mention varying moisture content in the burn material. Did the authors observe whether the wetter grasses produced more VOC (lower combustion efficiency) in any systematic way?

Figure 4: In general, the scale mismatch on the O3 timeseries makes immediate comparison between methods difficult. The authors should perhaps switch to a log-scale on the y-axis that can effectively compare low and high concentrations and offsets in both smoke plumes and background air. The authors attempt to explain the positive offset of the UV-C method outside of the burning period, but there is also a significant negative offset in the UV-C-H method that is not discussed. Could the authors provide more insight on why the UV-C-H and NO-CL techniques disagree in background air?

L378+: If the damaged MnO2 scrubber ineffectively removed O3, I would expect the

UV-C measurement to be biased low in background air rather than high. Please elaborate on the mechanism of MnO2 damage resulting in a significant positive offset. Also, it's unclear when the scrubber damage became an issue. Did it affect data from the 2017 prescribed burns?

Figure S9 indicates there is potential artifact even <1-2 ppm CO. Do these plots just use data from the burn periods or include points when chamber is flushed with outside air?

L459-461: How does the residence time and sample rate vary for each instrument?

Table 4: The slope and intercept uncertainties should be included with the fit parameters. How different are the range of fitted slope values statistically? In general, there is lack of uncertainty treatment in the paper. How do the uncertainties compare for each measurement technique? This information should be included in the manuscript.

L550-552: See question 1 above. How close to the plume do you have to be for interferences to matter? Is this relevant for air quality monitoring stations not located in the immediate vicinity of the fire line?

L554: What is estimated CO-$\Delta$O3 correlation for the chamber studies? It would still be worthwhile to include this information in the supplement.

Figures S9 and S10: Can you demonstrably separate CO-$\Delta$O3 regimes based on "burn condition"? The authors allude to this in the text (L563) and show an individual burn in Fig S10, but a more in-depth analysis of the contributing burn condition factors would provide a more quantitative and perhaps predictive assessment of how CO links to O3 artifacts under the varied burn conditions. The authors also perform separate regressions for NO2 and THC, but a separate correlation with humidity might be illustrative (if the data exists).

L571: Is it possible that interactions between water vapor and VOC somehow compound the VOC effect? In other studies (e.g., Spicer et al. 2010, Turnipseed et al.

[Figure]

2017), Nafion alone seems to play little role in mitigating VOC artifacts but does significantly reduce water vapor artifacts. In drier environments, does adding Nafion affect the positive artifact magnitude? This would be more conclusive evidence that Nafion does in fact remove certain permeable VOC species.

L605: Could this also be confounded by the faulty $MnO_2$ scrubber?

Technical Corrections: Table 1: Add uncertainty associated with each measurement technique. Sample rate would also be useful.

Figure S1 and other timeseries in general: It's difficult to compare NO-CL and UV measurements of plumes and background air given the large mis-match in scale. Some other way of presenting this material (e.g., semi-log) might help the visual comparison. The lines are also not very easy to distinguish. Using different colors instead of just patterns would help.

Figure 2. Does not need to be in 3D and could use a color scheme instead of patterns.

---

## Author Response (AR1)

**AMT-2020-383 - Authors Response**

**Reviewer 1 Comments – Responsenses and Manuscript Revisions**

**General comments.**

This paper uses measurements in highly concentrated fire plumes (within 100m of wildland grass fires,
and in controlled burns at the Missoula Fire Lab) to assess interferences in UV absorption measurements
of ozone at 254 nm.

This paper is motivated by the ozone measurements of UV absorption instruments, and the health impacts
from that ozone. Large increases in ozone may be observed after precursor NOx and VOC have had time
to react. The time scale to produce that ozone is highly dependent on plume dilution, which itself is highly
variable in time, but typically takes place over hours since emission. A fundamental question: for direct
emissions, the interfering species will also be diluted, such that the lowest interferences may be expected
at the highest levels of plume ozone. Secondary production of UV-active hydrocarbons, e.g., production
of nitroaromatics following oxidation in the presence of NO2, may dominate the ozone interference
downwind. What balance of directly emitted vs. secondary species are conjectured to lead to interferences
in ambient ozone measurements? Regardless of the source of the interference (primary vs. secondary),
given the lack of consistency from fire to fire (or even between different implementations of the UV
absorption technique) in the level of interferences measured, can the authors say what level of "fire
impact" causes a non-negligible interference? Is 1 ppm of ozone acceptable? 10 ppm of ozone?
Regardless, despite the experimental detail in this paper, it is not clear what ozone monitoring locations
are expected to suffer from significant interferences as a result of wildfires or prescribed burns. Lacking
these considerations, the paper's conclusions are qualitative at best, and by implication condemn a much
larger portion of the U.S. ozone monitoring network during the fire season than I suspect is warranted.
For a given UV absorption monitor, can they recommend what data to retain, and what data to eliminate
because of fire impacts? Some additional clarity in the real-world effects of fire smoke on ozone
monitoring is needed for this to make a novel and useful contribution to the literature.

The paper is overly long and can be shortened by removing extraneous details, repetitive text, and tables
that do not provide any usefully generalizable data as suggested below. Earlier literature is not well cited,
and additional references are also suggested below.

**Response:** The authors appreciate the time required to provide the review and feel that the suggestions
provided by the reviewer will result in an improved manuscript for resubmission.

The authors do agree that a more detailed look at data collected at sites being impacted by aged smoke
(ex. State and local regulatory monitoring sites being impacted by nearby wildfires and long range
transport of photochemicaly aged smoke plumes) and are currently collecting this data as part of the EPA
MASIC study in Boise, ID; Missoula, MT; and Reno, NV.  This additional data collection will aid in
linking these research chamber and near field prescribed grassland burn measurements back to real world regulatory monitoring situations. We will address these issues in a new "implications" section prior to
the manuscript conclusion.

The authors will, as suggested by the reviewer, attempt to shorten the length of the manuscript by
removing text that is tangential to the scope of the paper. In addition, the authors will review earlier
literature and cite as appropriate including those references suggested by the reviewer.

**Manuscript Revision:** The authors, as suggested by the reviewer removed text from the manuscript to
shorten the length of the document. An additional "Implications" section was added prior to the
conclusions section to tie the results of the research detailed in this manuscript to real world monitoring
applications. Included in this section will be a review of data from monitoring sites downwind of fires to
show the impact of the measurement artifacts described in this manuscript.

**Specific comments.**

**line 14:** "... large increases in ozone are also observed downwind ..." Is this always true?

**Response:** The authors did not imply that large increases in ozone are always observed down wind of
wildfire events. To clarify this and prevent assumptions that these increases in ozone "always occur", the
text will be rewritten to include a statement like "... large increases in ozone have been observed
downwind ..."

**Manuscript Revision:** text changed to read "... large increases in ozone have been observed downwind
..."

**line 32 (and lines 38 and 182 and elsewhere):** The NO-induced chemiluminescence measurement of
ozone is repeatedly described as "interference-free", which is misleading - it has a known dependency on
water vapor, which can lead to sensitivity variations of up to 8% if not accounted for. Please rephrase.

**Response:** The authors will rephrase these statements to emphasize that sample treatment steps, including
the use of a drier, must be taken prior to analysis to remove the effects of water vapor.

**Manuscript Revision:** Removed "interference free" from line 32. Removed "interference free" from
line 38. Clarifying statement inserted in line 62 "Both the ET-CL and NO-CL methods are subjet to slight
interfernces by water vapor. Howver, these potential interfenrces can be elimitated throught the use of
Nafion based drier or equivalent sample water vapor treatment system." Removed "interference free"
from line 182.

**line 52:** for clarity, please change to "... generates nitrogen dioxide in an electronically excited state..."
The original citation is Clough and Thrush, 1966, Chemical Communications,728, pp. 783-784.

**Response:** The authors agree with this suggestion and will change the text accordingly.

**Manuscript Revision:** Text changed to read "... generates nitrogen dioxide in an electronically excited
state..."

**line 93:** please remove CO2, as its absorption is negligible at 254 nm.

**Response:** The authors agree with this suggestion and will remove CO2 from the text accordingly.

**Manuscript Revision:** CO2 removed from the text

**line 142:** "...a supply of NO gas..." is not always needed - line 213 refers to one implementation of the
"scrubberless" UV absorption method uses a supply of N2O gas and produces NO by photolysis.

**Response:** The authors agree with this suggestion and to clarify will rewrite the sentence to read: "The
SL-UV method requires a continuous supply of compressed NO or nitrous oxide ($N_2O$) (which the
instrument converts to NO) to serve as the scrubber gas.

**Manuscript Revision:** Sentence rewritten to read "Similar to NO-CL, the SL-UV method requires a
continuous supply of compressed NO or nitrous oxide ($N_2O$) (which the instrument converts to NO) to
serve as the scrubber gas.

**lines 230 - 237:** Details of power, generator, charger, and batteries are tangential to the performance of
the analyzers and could be eliminated to shorten the text.

**Response:** The authors agree with this suggestion and will review the manuscript and eliminate non-
relevant text that will shorten the document.

**Manuscript Revision:** Details of power, generator, charger, and batteries and other non-relevant material
removed from the manuscript text.

**lines 267-8:** "...calibrations for THC were performed using... a methane/propane gas cylinder..." This
work eventually concludes that VOCs are "likely to interfere with UV absorption measurements of O3",
no surprise there. What is surprising is the rudimentary approach to quantifying those VOCs in this
manuscript. FID response factors vary with carbon number (for example, by up to a factor of 3 between
methane and propane!), between aliphatic, aromatic, and cyclic structures, and with heteroatomic
functionality. A sentence noting the uncertainty introduced in their measurement of VOCs (here called
THC) by using only methane and propane to determine FID sensitivity would be appropriate here.

**Response:** The authors agree with this comment and will add a sentence to address the uncertainty
associated with our use of the THC method and its calibration procedure to approximate VOC
concentrations.

**Manuscript Revision:** The THC calibration text was rewritten as follows to emphasize that the THC
results are an approximation of THC concentration in smoke "Per the manfactuerer provided operators
manual, calibrations for THC were performed using the T700U calibrator and a certified EPA
methane/propane gas cylinder (Airgas). FID response factors for organic compounds can vary
significantly based upon factors such as carbon number and compound class (Tong and Karasek 1984).
The carbon numbers for methane and propane vary by a factor of three and the FID response factors for
those compounds may also vary by a similar amount. In addition, the complex mixture of hydrocarbons
found in smoke will have large variations in carbon number and FID response factors. As such, the results
obtained with the THC analyzer are an approximation of THC (and VOC) concentrations in smoke. In addition, for THC calibrations, the T701H zero air generator was replaced with scientific grade zero air
compressed gas cylinders (Airgas)."

**Figure 2:** This is not a good graphic. There is absolutely no information conveyed by the third dimension
of this graph; please turn this into a 2D bar graph and improve the legibility of the different hatches. The
high level of interference from the UV-C and UVC- H techniques overwhelms any useful information on
the other techniques - suggest plotting only to 50 ppb and annotating the UV-C maxima with text. These
data are presented as O3 in ppb - what is the correct, or expectation value? The NO-CL data are lost in
this presentation and should be emphasized as the correct value.

**Response:** The authors agree the reviewers comment. The figure will be reformatted into 2D and
assuming that AMT allows colored figures will include a color scheme to improve clarity and viewability.
In addition, the y axis scale will be reduced to 50 ppb and the average values for all methods will be
included in the figure as text. The figure caption will be revised to reflect these changes.

**Manuscript Revision:** Figure 2 was reformatted into 2D and a color scheme added to improve
viewability. The y-axis scale was capped at 50 ppb and the average values for all methods and study
periods were included as text in the figure.

**Figure 4:** The NO-CL reference trace in the upper figure is the hardest to see; these figures could use
some work for legibility. The text refers to positive artifacts for the UV methods during burning periods,
ascribed to interferences from VOCs and PM2.5. Another problematic feature is the negative artifact
when the chamber is flushed with outside air, where the UV-C method falls below the NO-CL method
(bottom panel). Why is that? Did I miss the explanation?

**Response:** The authors will work on this time series as well as others to make the figures more legible
including looking into using a different scale on the y-axis. The post burn calibration checks on April 23,
2018 revealed a +8 % bias in the NO-CL method and a -2 % bias in the UV-C-H method. These biases
were evident during the chamber flush periods on that day. Each analyzer was re-zeroed and spanned
resulting in the elimination of the bias between the two methods as observed in the results from the
subsequesn day (April 24, 2018). This will be addressed in the figure caption.

**Manuscript Revision:** Figure 4 was reformatted to include a logarithmic scale for O3 concentrations
making comparisons between the different methods more clear. The following text was added to the figure
caption to address the bias observed during the chamber flush periods "The post burn calibration checks
on April 23, 2018 revealed a +8 % bias in the NO-CL method and a -2 % bias in the UV-C-H method.
These biases were evident during the chamber flush periods on that day. Each analyzer was re-zeroed and
spanned resulting in the elimination of the bias between the two methods as observed in the results from
the subsequesn day (April 24, 2018)."

**Lines 378-388:** I could not follow the confusing thread discussing how and when the MnO2 scrubber
failed in these experiments - for clarity I'd recommend deleting this section and removing all data taken
with an inoperative scrubber.

**Response:** The scope of this paper is a comparison/evaluation of ozone monitoring methods in smoke and the damage to the converter occurred while operating the UV-C analyzer in heavy smoke, the authors feel that this potential measurement issue is very important to those utilizing these instruments and should at a minimum be mentioned in this manuscript. The converter issue is important in that the effect continuous long after the smoke exposure is over and is not obvious when conducting typical QA/QC reviews (e.g., zero/span calibrations and checks). The authors will add/remove text to clarify when the damage occurred and the impact that the damaged converter had on the results obtained with the UV-C method.

**Manuscript Revision:** The authors clarified some text in this section but feel the section is well explained as to when the damage occurred and the overall impact. The section now reads "During the 2018 chamber burns the UV-C results were biased high by 15-20 ppb even during non-burn (i.e., overnight) periods as evident in Fig. 4 (top panel) and Fig. S4. The initial hypothesis was that the bias was associated with high chamber backgrounds of interfering species due to years of heavy burning in the chamber. However, it was later discovered during a subsequent summer/fall 2018 ambient air study in North Carolina in the absence of smoke, that sampling heavy smoke plumes during the fall 2017 prescribed grassland burns irreversibly damaged the $MnO_2$ scrubber in the UV-C instrument. The effect of the bias was observed mainly when sampling ambient air and not readily observed during routine calibration checks (zeroes and spans) except for an increase in the time required to obtain stable zero and span values. During the summer/fall 2018 North Carolina study and prior to the start of the 2019 chamber burns, a new $MnO_2$ scrubber was installed and resulted in a significant and immediate reduction of the observed high bias, shown in Fig. 4 (bottom panel) and Fig. S5."

**Table 3:** Since there appears to be very large fire-to-fire and technique-to-technique variability in the interferences, with no consistent dependence on any of the variables measured, quantifying their precise values in a table seems not very useful. I'm not sure what information this table provides; what quantitative use is it? Recommend deleting.

**Response:** The authors disagree with this comment. Regardless of the burning conditions or techniques used, artifacts in the UV photometric methods were observed and are presented in this table. The authors intend to include Table 3 in the manuscript.

**Manuscript Revision:** None

**line 498:** This section recommends using Nafion dryers to minimize smoke interferences in UV absorption ozone measurements. This begs the question - under what range of conditions does the use of a Nafion dryer allow EPA to actually accept an ozone measurement by the UV absorption measurement? Please discuss.

**Response:** This comment goes beyond the scope of this paper which is primarily focused evaluation/comparison of ozone monitoring methods in smoke plumes. However, the authors intend to include an additional implication section that will discuss the potential impact of our findings on real world monitoring application at sites that might be impacted by nearby wildfire smoke plumes.

**Manuscript Revision:** An implication section was added immediately preceding the conclusion section
that discusses the potential impact of our findings on real world monitoring application at sites that might
be impacted by nearby wildfire smoke plumes.

**Table 4:** Same comment as for Table 3, above: "Since there appears to be very large fire-to-fire and
technique-to-technique variability in the interferences, with no consistent dependence on any of the
variables measured, quantifying their precise values in a table seems not very useful. I'm not sure what
information this table provides; what quantitative use is it? Recommend deleting."

**Response:** The authors disagree with this comment. Regardless of the burning conditions or techniques
used, artifacts in the UV photometric method were observed and those artifacts are correlated with makers
of combustion as illustrated in this table. The authors intend to include Table 4 in the manuscript.

**Manuscript Revision:** None

**line 581:** I would suggest the authors review and cite the use of perfluorosulfonate membrane tubing to
remove UV-active hydrocarbons, e.g., in SO2 pulsed fluorescence instruments (Luke, W., 1997, JGR,
102, 16,255-16,265).

**Response:** The authors will review the suggested manuscript and if appropriate cite in the text as a
possible solution in mitigating interferences by wildfire generated UV-active hydrocarbons as suggested
by the reviewer.

**Manuscript Revision:** None. The authors reviewed the suggested manuscript and choose not to cite it
in this manuscript. The authors could not find mention of perfluorosulfonate membrane in the manuscript
which is similar to the make up of Nafion but did notice several instances of the proprietary "kicker" that
may or may not remove interfering hydrocarbons.

    **Reviewer 2 Comments – Responsenses and Manuscript Revisions**
**General Comments:** .

This study compares O3 measurement techniques in fresh, concentrate smoke plumes. The authors sample
smoke plumes from both prescribed prairie grass burns and controlled chamber burns using a NO
chemiluminescence measurement as the interference-free standard with which to compare several
iterations of UV absorption-based measurements. This study is motivated by the prevalence of UV-based
O3 analyzers at EPA air quality monitoring stations and the increasing impact of fire emissions on local
and regional air quality. Although these comparisons provide insight into the potential for UV-active
VOCs in smoke plumes to generate positive artifacts in the UV-based O3 measurements, a more
quantitative assessment is limited by the lack of detailed VOC measurements and the inability to
quantitatively disentangle the various CO-O3 regimes. The authors also suggest the role of Nafion in
mitigating potential artifacts, but do not provide enough information on the relative humidity conditions
during the various sampling periods or the potential for interactions between water vapor and VOC.
Further, the analysis emphasizes the effects of VOC interreferences in near-fire smoke plumes but does not provide much discussion on how the potential for interference diminishes with plume age and
dispersion. For example, how quickly do VOC react/diffuse to the point where their levels are no longer
of concern? How many ozone monitoring sites would be practically affected by these interferences?

**Response:** The authors appreciate the time required to provide the review and feel that the suggestions
provided by the reviewer will result in an improved manuscript for resubmission.

During both the prescribed and chamber burns, data were obtained for RH values and water vapor
concentration and is included in the data associated with this paper that will be provided through the EPA
Science Hub Web site (https://catalog.data.gov/dataset/epa-sciencehub) following the acceptance of this
paper. .   However the correlations between RH and the magnitude of the ozone artifact were not
significant and therefore not included in the manuscript.  In general, both the prescribed fire and chamber
burns were conducted under dry conditions with RH≤50%.  Past studies, which are now referenced in the
updated manusript indicate that at those RH values humidity effects are expected to have little to no
impact.  It is the intention of the authors to add an additonal section to this manuscript discussing
implications of this research on real world ozone monitoring such as that that occurs at State and local
moitoing sites.  The authors intend to review data from sites downwind of wildfires that potentialy show
the artifact in the UV-C O3 method and how it is correlated with markers of combustion processes.  As
stated in the text of the manuscript, the authors plan future studies to dig deeper into the hypothesized
VOC caused artifact and which will include, as the reviewer suggest looking into interaction between
VOCs and water vapor and the capabilities of Nafion in removing certain VOCs.

**Manuscript Revision:** An additional "Implications" section was added prior to the conclusions section
to tie the results of the research detailed in this manuscript to real world monitoring applications.  Included
in this section will be a review of data from monitoring sites downwind of fires to show the impact of the
measurement artifacts described in this manuscript.

**Specific Comments:**

**L243-244:** Is there any dependence of the artifact magnitude on distance from the active fire line? How
quickly do the VOC react/diffuse to the point where their levels are no longer detectable as a positive
artifact? All the measurements presented are taken within ~100 m from the fires, but any data collected
from aged smoke would be a useful counterpoint.

**Response:** The authors did not look at the dependencies of the artifact magnitude on distance from the
active fire line.  However, the authors do agree that a more detailed look at data collected at sites being
impacted by aged smoke (ex. State and local monitoring sites being impacted by nearby wildfires).  This
would aid in tying these measurments made in or near plume back to real world monitoring situations.
Most likely this will be done by adding an implications section prior to the manuscript conclusion.

**Manuscript Revision:** An implications section was added prior to the conclusion to address some of
reviewer 2 comments.

**L262:** The authors mention a +/- 10% performance objective between analyzers. Do the calibrations reveal any systematic offset between the CL and UV analyzers? In describing the prescribed and chamber burns, the authors mention varying moisture content in the burn material. Did the authors observe whether the wetter grasses produced more VOC (lower combustion efficiency) in any systematic way?

**Response:** The calibrations only revealed a significant offset during one period during this study. The post burn calibration checks on April 23, 2018 revealed a +8 % bias in the NO-CL method and a -2 % bias in the UV-C-H method. These biases were evident during the chamber flush periods on that day. Each analyzer was re-zeroed and spanned resulting in the elimination of the bias between the two methods as observed in the results from the subsequesn day (April 24, 2018). All other calibrations did not reveal any systematic offsets or biases between the different analyzers and we will clarify this in the updated version of the manuscript. At present the authors have not investigated the relationship between fuel moisture content and VOC production. In order to simulate a range of natural burning conditions, the chamber burns manipulated the moisture content, fuel type (pine needles, pine needles + fine woody debris), and bulk density of the fuelbeds. These fuelbed properties influence the relative mix of flaming and smoldering combustion and the chamber burns covered a range of combustion efficiencies (modified combustion efficiencies of 0.85 – 0.97). The authors will investigate further and address these findings in a future manuscript.

**Manuscript Revision:** The following text was added to the figure caption to address the bias observed during the chamber flush periods "The post burn calibration checks on April 23, 2018 revealed a +8 % bias in the NO-CL method and a -2 % bias in the UV-C-H method. These biases were evident during the chamber flush periods on that day. Each analyzer was re-zeroed and spanned resulting in the elimination of the bias between the two methods as observed in the results from the subsequesn day (April 24, 2018)."

The following text was also added to section 3.2 "The post burn calibration checks on April 23, 2018 revealed a +8 % bias in the NO-CL method and a -2 % bias in the UV-C-H method. These biases were evident during the chamber flush periods on that day. Each analyzer was re-zeroed and spanned resulting in the elimination of the bias between the two methods as observed in the results from the subsequesn day (April 24, 2018)." No other calibration corrections werer made during the 2018 and 2019 chamber studies."

**Figure 4:** In general, the scale mismatch on the O3 timeseries makes immediate comparison between methods difficult. The authors should perhaps switch to a log-scale on the y-axis that can effectively compare low and high concentrations and offsets in both smoke plumes and background air. The authors attempt to explain the positive offset of the UV-C method outside of the burning period, but there is also a significant negative offset in the UV-C-H method that is not discussed. Could the authors provide more insight on why the UV-C-H and NO-CL techniques disagree in background air?

**Response:** The authors will work on this time series plot as well as others to make the figures more legible including looking into using a different scale on the y-axis. As suggested by the reviewer, the authors will provide more insight into why the UV-C-H and NO-CL techniques disagree in background air.

**Manuscript Revision:** Figure 4 was reformatted to include a logarithmic scale for O3 concentrations making comparisons between the different methods more clear.

**L378+:** If the damaged MnO2 scrubber ineffectively removed O3, I would expect the UV-C measurement to be biased low in background air rather than high. Please elaborate on the mechanism of MnO2 damage resulting in a significant positive offset. Also, it's unclear when the scrubber damage became an issue. Did it affect data from the 2017 prescribed burns?

**Response:** In order for the scrubber to work correctly, it must remove O3 and only O3. Based upon the data, the damage most likely resulted in the scrubber also removing significant amounts of interfereing species during the reference measurement which would then be detected as ozone during the sample measurement resulting in the positive artifact. The data collected during the 2017 prescribed burns indicate that the scrubber was functioning properly in that there was excellent agreement between the UV-C and NO-CL methods when sampling out of the smoke plume.

**Manuscript Revision:** To clarify the section describing the bias observed during the 2018 chamber studies was re-written as follows: "During the 2018 chamber burns the UV-C results were biased high by 15-20 ppb even during non-burn (i.e., overnight) periods as evident in Fig. 4 (top panel) and Fig. S4. The initial hypothesis was that the bias was associated with high chamber backgrounds of interfering species due to years of heavy burning in the chamber. However, it was later discovered during a subsequent summer/fall 2018 ambient air study in North Carolina in the absence of smoke, that sampling heavy smoke plumes during the fall 2017 prescribed grassland burns followed by subsequent storage of the UV-C analyzer, irreversibly damaged the $MnO_2$ scrubber in the UV-C instrument. It is hypothesized that the damage resulted in the scrubber removing some of the interfering species in additon to ozone, preventing them from being removed in the reference measurment, and subsequent detection as ozone (positive bias) during the measurment cycle. The effect of the bias was observed mainly when sampling ambient/chamber air and not readily observed during routine calibration checks (zeroes and spans) except for an increase in the time required to obtain stable zero and span values. The bias was not observed during any of the 2017 prescribed grassland burns. During the summer/fall 2018 North Carolina study and prior to the start of the 2019 chamber burns, a new $MnO_2$ scrubber was installed and resulted in a significant and immediate reduction of the observed high bias, shown in Fig. 4 (bottom panel) and Fig. S5."

**Figure S9** indicates there is potential artifact even <1-2 ppm CO. Do these plots just use data from the burn periods or include points when chamber is flushed with outside air?

**Response:** Figure S9 includes data from the burn periods only. In the figure caption it describes it as "in-plume". The authors will add clarifying text similar to the following, "…and THC for all in-plume (burn period only) measurements…".

**Manuscript Revision:** The figure caption was re-written as follows: "Scatter plots between FRM and FEM $O_3$ differences and CO, $NO_2$, and THC for all in-plume (burn period only) measurements made during the 2018 and 2019 Missoula Fire Chamber studies. Observation points have been colored by the $O_3$ instrument. Over all observations there is little correlation between the $O_3$ instrument differences, but straight line structures within the overall scatters indicate that individual burn events measured in
the chamber have good correlations with distinct ratios."

**L459-461:** How does the residence time and sample rate vary for each instrument?

**Response:** Sampling rates and hence residence times are going to be similar for all instruments as they
all operate with similar flow rates. The authors will address this comment by either adding analyzer flow
rate to Table 1 or by inserting text in the Methods section under each corresponding analyzer type.
Generally, UV photometric type analyzers require a greater flow rate becaust the flow is split between
the two cells (refernce and measurment). The NO-CL method has only a single cell and requires a much
smaller flow rate to achieve a similar residence time.

**Manuscript Revision:** The flow rates of each method along with manufacturer reported performance
specifications were included in Table S1 which was added to the supplementary materials document. In
the text describing each method, a sentence similar to the following was added "Manufacturer provided
performance specifications for the NO-CL based TAPI T265 are given in Table S1."

**Table 4:** The slope and intercept uncertainties should be included with the fit parameters. How different
are the range of fitted slope values statistically? In general, there is lack of uncertainty treatment in the
paper. How do the uncertainties compare for each measurement technique? This information should be
included in the manuscript.

**Response:** The authors agree with this comment and will work to include uncertainties (both in tables
and in the text) of measurement methods and in fit parameters associated with regression statistics.

**Manuscript Revision:** Data for the Konza March 2017 were re-analyzed and new values included fro
slope, intercept R2 and n. The previous analysis included a few values that were associated with CO
levels that were below 1 ppm (our threshold of sampling in plume). Standard errors for the regression
slope and intercept were included in Table 4. In addition, the following text was added to discuss the
results of the regression analysis between markers of combustion CO and THC and the magnitude of the
ozone artifact: "The slight differences in the magnitude of the artifacts (fitted regression slopes) along
with the low uncertainty (standard errors) values indicate that the magnitude of the artifact may be
influenced by local conditions that make each burn unique. Such conditions might include meteorological
conditions, fuel composition, fuel moisture content, and times spent in combustion phase (flaming vs
smoldering)."

**L550-552:** See question 1 above. How close to the plume do you have to be for interferences to matter?
Is this relevant for air quality monitoring stations not located inthe immediate vicinity of the fire line?

**Response:** The authors focused on determining if significant ozone measurement artifacts do occur in
near-field smoke events and did not look at the dependencies of the artifact magnitude as a function of
distance from the active fire line. However the authors do agree that a more detailed look at data collected
at sites being impacted by aged smoke (ex. State and local monitoring sites being impacted by nearby
wildfires) and are currently collecting this data as part of the EPA MASIC study in Boise, ID; Missoula,
MT; and Reno, NV. This additional data collection will aid in linking these research chamber and near field prescribed grassland burn measurements back to real world regulatory monitoring situations. We
will address these issues in a new "implications" section prior to the manuscript conclusion.

**Manuscript Revision:** An implications section was added to the manuscript prior to the conclusion to
address this and other comments provided by reviewer 2.

**L554:** What is estimated CO-$\Delta$O3 correlation for the chamber studies? It would still be worthwhile to
include this information in the supplement.

**Response:** Regarding the correlation between $\Delta$O3 and CO from the chamber based burns, the authors
refer the reviewer to the original manuscript text:

"As indicated, $\Delta$O3(UV-C) and CO appear to be correlated in time but when performing linear regression
comparisons of $\Delta$O3(UV-C) and CO during each years chamber burns as a whole, correlations tend to be
poor. We suspect the positive O3 bias is driven by one or more VOCs (likely oxygenated VOCs). In fresh
smoke the excess concentrations of individual VOCs ($\Delta$X), and VOC sums ($\Delta$VOC), tend to be highly
correlated with $\Delta$CO (Yokelson et al., 1999; Gilman et al. 2015). The emission ratios of individual VOCs
to CO ($\Delta$X/$\Delta$CO) can vary considerably with combustion conditions such as fuel type and condition (e.g.
moisture content and decay state), fuel bed properties, such as bulk density, and the relative mix of
flaming and smoldering combustion (Gilman et al. 2015; Koss et al., 2017). Additionally, the response
of $\Delta$X/$\Delta$CO to burn conditions varies among VOCs. When each burn is considered individually or in
groups with similar conditions, the correlations between $\Delta$O3, CO, and THC are enhanced. An example
of this behavior is shown in Supplementary Fig. S10."

With that being stated, the authors will consider adding the CO-$\Delta$O3 correlation (both for the entire
chamber study period and also a subset of individual burns) either in Table 4 or in the body of the text
give evidence to the above statement. Visual representations of the correlations are given in Figures S9
and S10.

**Manuscript Revision:** The following text was added to section 3.4 to address this comment: "For the
chamber burns the magnitude of the ozone artifacts in ppb apparent $O_3$ per ppm CO, ranges between 6 -
210 ppb ppm$^{-1}$ for the individual burns. $R^2$ and standard error values were consistent with those observed
dring the prescribed burns (see Table 4). " In addition, the requested information is provided visually in
figures S9 and S10.

**Figures S9 and S10:** Can you demonstrably separate CO-$\Delta$O3 regimes based on "burn condition"? The
authors allude to this in the text (L563) and show an individual burn in Fig S10, but a more in-depth
analysis of the contributing burn condition factors would provide a more quantitative and perhaps
predictive assessment of how CO links to O3 artifacts under the varied burn conditions. The authors also
perform separate regressions for NO2 and THC, but a separate correlation with humidity might be
illustrative (if the data exists).

**Response:** The authors will consider elaborating further per the reviewers suggestion on CO-$\Delta$O3
regimes based on burn conditions (i.e., individual burns or burns grouped by similar burn conditions).
The authors previously attempted to establish a correlation between $\Delta$O3 and humidity (water vapor concentration) but those correlation were extremely poor. As such the authors chose not to include this
analysis.

**Manuscript Revision:** The following text was added to section 3.4 to elaborate on the lack of correlation
between $\Delta O3$ and CO when considered as a whole but showing improvements when considering
individual burns: "For the chamber burns the magnitude of the ozone artifacts in ppb apparent $O_3$ per ppm
CO, ranges between 6 - 210 ppb ppm$^{-1}$ for the individual burns. $R^2$ and standard error values were
consistent with those observed dring the prescribed burns (see Table 4)."

**L571:** Is it possible that interactions between water vapor and VOC somehow compound the VOC effect?
In other studies (e.g., Spicer et al. 2010, Turnipseed et al. 2017), Nafion alone seems to play little role in
mitigating VOC artifacts but does significantly reduce water vapor artifacts. In drier environments, does
adding Nafion affect the positive artifact magnitude? This would be more conclusive evidence that Nafion
does in fact remove certain permeable VOC species.

**Response:** Both the 2017 prescribed fire and 2018-2019 chamber based burns were conducted under dry
conditions (RH≤50%) and humidity interferences are expected to be minimal. As stated in the previous
comment, the correlation between in plume water vapor concentration and $\Delta O3$ was not significant. In
addition, there is no signifcant correlation between the magnitude of the artifact and RH. In both the
prescribed grassland and chamber burns there was a UV instrument with a Nafion drier and a UV
instrument without the drier and they were operated simultaneously. The magnitude of the artifact (both
average and maximum) was greatly reduced in the method using the Nafion drier. This is evident in
comparing the magnitude of the UV-C artifact with that of the UV-C-H (UV method employing a Nafion
based drying sytem. In all cases, the UV-C artifact was nearly an order of magnitude greater than that of
the UV-C-H. This is also became furher evdent when the Nafion drier was added to the UV-C method
on the final day of burning during the 2018 chamber studies, thus reducing the magnitude of the UV-C
artifact to a point comparible to that of the UV-C-H method. The effect of Nafion on the magniturde of
the artifact is detailed in section 3.3. In section 3.5 of the manuscript, the authors will attempt to clarify
that in addition to our hypothesis of certain VOCs being removed by the Nafion, there may also be
interactions between water vapor and VOCs that may be confounding the observed artifact.

**Manuscript Revision:** The authors feel that text and discussion provided in section 3.3 already provide
a response to the reviewer 2's comment suggestion. As stated in the response listed above, during this
study humidity effects are expected to be at a minimum due to the low RH values that existed during all
study periods. As such and to clarify, the following text was inserted in section 2.6: "In general, chamber
RH values were below 50% facilitating dry burning condition." And section 3.1: "In addition, ambient
RH values were generally belwo 50% indicatiibng that the spring and fall 2017 prescribed burns were
cunducted under dry conditions."

The last sentence of section 3.4 was re-written to read "Considering that the prescribed grassland and
chamber burns were conducted under dry conditions, the size of the difference (as large as hundreds of
ppb) cannot be explained purely by the previously observed relative humidity effects on measurements
(Leston et al., 2005; Wilson et al., 2006), suggesting that the Nafion® dryer is directly impacting the
concentrations of other interferents in the sample stream."

**L605:** Could this also be confounded by the faulty MnO2 scrubber?

**Response:** We do know that during the 2018 chamber studies the damaged scrubber did cause an
approximate +10-15 ppb bias in the UV-C method which was present even in the absence of smoke. At
the end of the 2018 chamber studies, the authors added a Nafion drier to the UV-C method as indicated
in Figure 4.  The addition of the Nafion to the UV-C method reduced the magnitude of the artifact by a
factor of three making it compatible to the artifact observed for the UV C-U method.  The addition of the
nafion did result in a slight reduction in the bias that we attributed to damaged scrubber but not on the
order of 3X.  W suspect that the addition of the drier would reduce or remove many of the VOC species
prior to also being removed by the faulty scrubber thus resulting in a reduction of the bias but not
completely eliminating it. The authors will add clarifying text in the body of the manuscript explaining
the damage to the MNO2 scrubber and its hypothesized effect on the oberved bias.  The reviewers
comment would only apply to the 2018 chamber study as the MnO2 scrubber in the UV-C method was
functioning properly during all other studies.

**Manuscript Revision:** Clarifying text was added in section 3.2 to explain the effect that the damaged
scrubber had on the UV-C ozone results (positive bias).

**Technical Corrections:**

**Table 1:** Add uncertainty associated with each measurement technique. Sample rate would also be useful.

**Response:** The authors will address this comment by either adding analyzer flow rate and uncertainties
to table 1 or by inserting text in the Methods section under each corresponding analyzer type.

**Manuscript Revision:** An additional table (Table S1) was added to the supplemental materials document
containing manufacturer provided performance specifications for each analyzer to address this comment
from reviewer 2.  In the text describing each method, a sentence similar to the following was added
"Manufacturer provided performance specifications for the NO-CL based TAPI T265 are given in Table
S1.

**Figure S1 and other timeseries in general:** It's difficult to compare NO-CL and UV measurements of
plumes and background air given the large mis-match in scale. Some other way of presenting this material
(e.g., semi-log) might help the visual comparison.The lines are also not very easy to distinguish. Using
different colors instead of just patterns would help.

**Response:** The authors agree with this comment and will take steps to improve the  the time series plots,
including looking into different scales (e.g. semi-log) and also using colored lines in the figures.

**Manuscript Revision:** Figures 4 and S1-5 were reformatted adding logarithmic scales where appropriate
and color schemes to improve readability.

**Figure 2:** Does not need to be in 3D and could use a color scheme instead of patterns.

**Response:** The authors agree with the reviewers comment. The figure will be reformatted into 2D and
assuming that AMT allows colored figures will include a color scheme to improve clarity and view ability.
In addition, the y axis scale will be reduced to 50 ppb and the average values for all methods will be
included in the figure as text. The figure caption will be revised to reflect these changes.

**Manuscript Revision:** Figure 2 was reformatted into 2D and a color scheme added to improve
viewability. The y-axis scale was capped at 50 ppb and the average values for all methods and study
periods were included as text in the figure.